# Selective posttranslational inhibition of Ca_Vβ_1-associated voltage-dependent calcium channels with a functionalized nanobody

Travis J. Morgenstern[1], Neha Nirwan[2], Erick O. Hernández-Ochoa[3], Hugo Bibollet [3], Papiya Choudhury[4], Yianni D. Laloudakis[5], Manu Ben Johny [4], Roger A. Bannister[3,6], Martin F. Schneider[3], Daniel L. Minor Jr [2,7,8,9,10,11] & Henry M. Colecraft [1,4] ✉

Ca$^{2+}$ influx through high-voltage-activated calcium channels (HVACCs) controls diverse cellular functions. A critical feature enabling a singular signal, Ca$^{2+}$ influx, to mediate disparate functions is diversity of HVACC pore-forming α$_1$ and auxiliary Ca$_V$β$_1$–Ca$_V$β$_4$ subunits. Selective Ca$_V$α$_1$ blockers have enabled deciphering their unique physiological roles. By contrast, the capacity to post-translationally inhibit HVACCs based on Ca$_V$β isoform is non-existent. Conventional gene knockout/shRNA approaches do not adequately address this deficit owing to subunit reshuffling and partially overlapping functions of Ca$_V$β isoforms. Here, we identify a nanobody (nb.E8) that selectively binds Ca$_V$β$_1$ SH3 domain and inhibits Ca$_V$β$_1$-associated HVACCs by reducing channel surface density, decreasing open probability, and speeding inactivation. Functionalizing nb.E8 with Nedd4L HECT domain yielded Chisel-1 which eliminated current through Ca$_V$β$_1$-reconstituted Ca$_V$1/Ca$_V$2 and native Ca$_V$1.1 channels in skeletal muscle, strongly suppressed depolarization-evoked Ca$^{2+}$ influx and excitation-transcription coupling in hippocampal neurons, but was inert against Ca$_V$β$_2$-associated Ca$_V$1.2 in cardiomyocytes. The results introduce an original method for probing distinctive functions of ion channel auxiliary subunit isoforms, reveal additional dimensions of Ca$_V$β$_1$ signaling in neurons, and describe a genetically-encoded HVACC inhibitor with unique properties.

Ion channels are integral membrane protein complexes that control flux of ions into and out of cells and organelles to regulate essential physiological processes including neuronal firing and synaptic transmission, muscle contraction, hormonal secretion, fluid homeostasis, and gene expression[1]. Surface membrane ion channels are often multisubunit assemblies composed of pore-forming polypeptides assembled with auxiliary subunits. Within an ion channel class, functional diversification is frequently achieved by molecular diversity of both

[1]Department of Molecular Pharmacology and Therapeutics, Columbia University Irving Medical Center, New York, NY, USA. [2]Cardiovascular Research Institute, University of California, San Francisco, CA, USA. [3]Department of Biochemistry and Biology, University of Maryland School of Medicine, Baltimore, MD, USA. [4]Department of Physiology and Cellular Biophysics, Columbia University Irving Medical Center, New York, NY, USA. [5]Vagelos College of Physicians and Surgeons, Columbia University Irving Medical Center, New York, NY, USA. [6]Department of Pathology, University of Maryland School of Medicine, Baltimore, MD, USA. [7]Department of Biochemistry and Biophysics, University of California, San Francisco, CA, USA. [8]Department of Cellular and Molecular Pharmacology, University of California, San Francisco, CA, USA. [9]California Institute for Quantitative Biomedical Research, University of California, San Francisco, CA, USA. [10]Kavli Institute for Fundamental Neuroscience, University of California, San Francisco, CA, USA. [11]Molecular Biophysics and Integrated Bio-imaging Division, Lawrence Berkeley National Laboratory, Berkeley, CA 94720, USA. ✉e-mail: hc2405@cumc.columbia.edu

pore-forming $\alpha_1$ and auxiliary subunits in the form of either subtypes encoded by distinct genes or splice variants. Selective inhibition of distinct ion channels is a vital capability that not only enables delineation of their specific physiological roles but also has provided therapeutics for many diseases. Posttranslational inhibition of ion channels is typically accomplished pharmacologically using small molecules or peptides that target pore-forming $\alpha_1$ subunits. By contrast, posttranslational inhibition of ion channels based on the identity of their auxiliary subunits is rare, particularly when these accessory proteins are cytosolic. This deficiency is a critical blind spot that hampers in-depth understanding of the functional significance of ion channel molecular diversity and limits opportunities for developing novel potential therapeutics.

High-voltage activated calcium channels (HVACCs) convert information encoded in electrical signals to $Ca^{2+}$ influx into cells to drive important biological responses including synaptic transmission in neurons and muscle contraction in the heart[2]. There are seven distinct HVACCs ($Ca_V1.1$-$Ca_V1.4$ $Ca_V2.1$- $Ca_V2.3$), classified based on the identity of the pore-forming $\alpha_1$ subunit ($\alpha_{1A}$- $\alpha_{1F}$; $\alpha_{1s}$), which contains the voltage sensor, selectivity filter, and channel pore. Mature HVACCs are multi-subunit complexes comprised of pore-forming $\alpha_1$ polypeptides assembled with auxiliary $\beta$, $\alpha_2\delta$, and $\gamma$ subunits. There are four $Ca_V\beta$ subunits isoforms ($Ca_V\beta_1$ - $Ca_V\beta_4$) encoded by distinct genes[3]. In most cell types, $Ca_V\beta$s are required for $\alpha_1$-subunit targeting to the plasma membrane[4–10]. Beyond surface density, $Ca_V\beta$s also regulate different aspects of channel gating including open probability ($P_o$), voltage-dependence of activation and inactivation, and inactivation kinetics[3,11–16].

HVACCs are important therapeutic targets for serious cardiovascular and neurological diseases[2,17]. There are selective small molecule and peptide toxin blockers that target HVACC pore-forming $\alpha_1$ subunits which have been critical in deciphering the distinctive physiological roles of $\alpha_1$ isoforms[2,18–20]. By contrast, there is no method currently available to inhibit HVACCs posttranslationally based on the identity of the $Ca_V\beta$ isoform associated with them. This limits the ability to discern the functional logic for $Ca_V\beta$ molecular diversity in tissues where multiple $\alpha_1$ and $\beta$-subunit isoforms are co-expressed. Here, we describe a nanobody, nb.E8, that exclusively binds auxiliary $Ca_V\beta_1$ subunits, and significantly inhibits whole-cell current through $Ca_V\beta_1$-associated HVACCs by decreasing channel surface density and $P_o$. Fusing Nedd4L HECT domain to nb.E8 generated a tool we have named Chisel-1 (calcium channel inhibitor via selective targeting of $Ca_V\beta_1$) that selectively abolishes $Ca_V\beta_1$-bound HVACC currents in heterologous and native cells. Deploying Chisel-1 in hippocampal neurons revealed a dominant contribution of $Ca_V\beta_1$-associated HVACCs to excitation-transcription coupling in neurons. Overall, this work demonstrates a method for developing auxiliary-subunit-isoform-selective posttranslational inhibitors of HVACCs that can be broadly applied to other ion channels and multi-subunit membrane protein complexes.

## Results
### Rationale for developing post-translational $Ca_V\beta$-isoform-selective HVACC inhibitors
The physiological roles of $Ca_V\beta$ isoforms have traditionally been probed using gene knockout or knockdown methods[21–25]. While such approaches have been invaluable in increasing understanding of the functional role of $Ca_V\beta$ molecular diversity in cells, they often lead to ambiguous results that can confound interpretation. Figure 1 shows two examples of how gene knockout/knockdown of $Ca_V\beta$ isoforms may yield results that are ambiguous to interpret. The first example stems from the observation that inducible knockout of $Ca_V\beta_2$ in adult mouse heart results in only a moderate impact on basal $Ca_V1.2$ current ($I_{Ca,L}$) amplitude[26] (Fig. 1a). This result indicated that in contrast to the dogma established by heterologous expression experiments, plasma

membrane targeting of $Ca_V1.2$ in adult heart cells was not absolutely dependent on association with a $Ca_V\beta$ subunit, leading to a quandary in interpretation. Is there a prominent subset of $Ca_V1.2$ channels in adult cardiomyocytes that are not complexed to $Ca_V\beta$? The import of this question was heightened by the finding that association with a $Ca_V\beta$ is necessary for $\beta$-adrenergic upregulation of $Ca_V1.2$ which is necessary for the physiologically crucial fight-or-flight response[27]. We developed a nanobody-based molecule that posttranslationally inhibits voltage-gated $Ca^{2+}$ channels by targeting auxiliary $Ca_V\beta$ subunits[28]. The molecule, termed $Ca_V$-a$\beta$lator, consists of a nanobody (nb.F3) that indiscriminately binds all four $Ca_V\beta$ isoforms, and is fused to the catalytic HECT domain of the E3 ubiquitin ligase, Nedd4L. $Ca_V$-a$\beta$lator potently inhibits $Ca_V1/Ca_V2$ channels by binding associated $Ca_V\beta$s and increasing ubiquitination of both $\alpha_1$ and $Ca_V\beta$ subunits thereby removing the channel complex from the cell surface. Expression of $Ca_V$-a$\beta$lator in cardiac myocytes eliminated $I_{Ca,L}$ by retaining $Ca_V1.2$ in intracellular organelles, primarily late endosomes[28], definitively revealing that essentially all $Ca_V1.2$ $\alpha_{1C}$ subunits are bound to $Ca_V\beta$ in adult ventricular cardiomyocytes (Fig. 1a).

A second scenario visualizes a situation in which two $Ca_V\beta$ isoforms (e.g. $Ca_V\beta_1$ and $Ca_V\beta_2$) preferentially associate with distinct $\alpha_1$ subunits to mediate different downstream functional effects (outputs 1 and 2, Fig. 1b) in a cell. Gene knockout or knockdown of $Ca_V\beta_1$ isoform in this scenario may not result in an elimination of functional output 1 if $Ca_V\beta_2$ can replace it and has an overlapping function. This circumstance would result in an erroneous conclusion that $Ca_V\beta_1$ is not involved in output 1 (Fig. 1b). By contrast, if it were possible to selectively inhibit $Ca_V\beta_1$-associated channels post-translationally this would reveal a more accurate picture of the functional organization of $Ca_V1$/$Ca_V2$ channels and the divergent functions enabled by $Ca_V\beta$ molecular diversity.

Given the success of $Ca_V$-a$\beta$lator as a $Ca_V\beta$-targeted potent posttranslational inhibitor of $Ca_V1/Ca_V2$ channels, we wondered whether it would be possible to develop $Ca_V\beta$-isoform-selective blockers using a similar targeted ubiquitination design principle. Accordingly, we aimed to identify a nanobody that by contrast to nb.F3 was selective for a particular $Ca_V\beta$ isoform.

### Isolation and characterization of a $Ca_V\beta_1$-selective nanobody
We previously immunized a llama with purified $Ca_V\beta_1$ and $Ca_V\beta_3$ subunits and generated a phagemid library of nanobodies (>$1 \times 10^7$ library size) from peripheral blood mononuclear cells. Subsequent phage display using $Ca_V\beta_1$ as bait yielded a number of positive nanobody binders as determined by enzyme linked immunosorbent assay (ELISA). Several clones that showed strong binding by ELISA were sequenced and cloned into mammalian expression vectors[28]. To develop a robust and relatively high throughput approach to evaluate the binding and selectivity of the isolated nanobodies for $Ca_V\beta$ isoforms we adapted a flow cytometry Förster resonance energy transfer (flow-FRET) method[29,30]. As a test of the approach, we first evaluated the binding of nb.F3, which we previously established binds all four $Ca_V\beta$ isoforms using a low throughput membrane co-translocation assay[28]. We tagged nb.F3 with Venus and $Ca_V\beta$ isoforms ($Ca_V\beta_1$ - $Ca_V\beta_4$) with Cerulean fluorescence proteins, respectively. HEK293 cells were transiently co-transfected with Venus-nb.F3 and Cerulean-$\beta$ or Cerulean alone as a negative control (Fig. 2a). In accord with our previous finding, Venus-nb.F3 showed strong FRET with all four Cerulean-tagged $Ca_V\beta$ isoforms (Fig. 2b). Scatter plots of FRET signal as a function of free Venus-nb.F3 (acceptor) concentration were fit with a 1:1 binding model yielding binding affinity estimates for all four $Ca_V\beta$s (Fig. 2b).

We applied the flow-FRET approach to screen for binding and potential selectivity of different nanobody clones for distinct $Ca_V\beta$ isoforms. One of these, nb.E8, displayed strong binding to $Ca_V\beta_1$ with an affinity similar to that of nb.F3, but essentially no association with

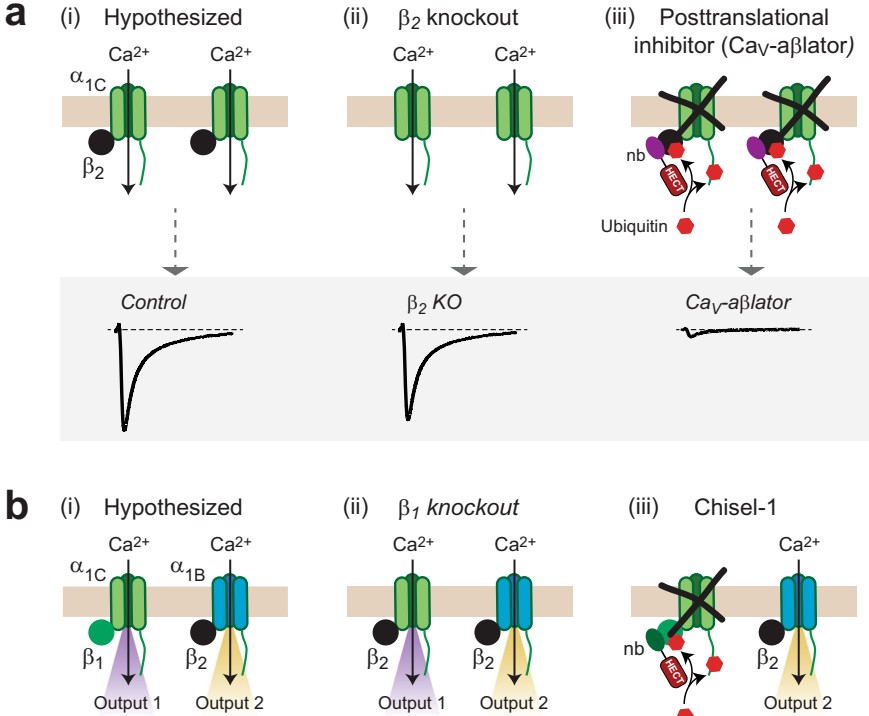

**Fig. 1 | Rationale for developing post-translational Ca$_V$β-dependent inhibitors for Ca$_V$1/Ca$_V$2 channels. a** Cartoon depicting hypothesized association of Ca$_V$1.2 with β$_{2b}$ subunits in adult ventricular myocytes (i) and the expected loss of β$_{2b}$ with gene knockout of this subunit (ii). Bottom, knock out of β$_2$ in adult heart has only minimal impact on whole-cell Ca$^{2+}$ current making it ambiguous whether most Ca$_V$1.2 are associated with β$_{2b}$ in adult ventricular cardiomyocytes. Posttranslational inhibition of cardiac Ca$_V$1.2 using a Ca$_V$β-targeted nanobody fused to Nedd4L HECT (Ca$_V$-aβlator) domain eliminates Ca$_V$1.2 complexes from the membrane and abolishes current (iii), proving that Ca$_V$1.2 is stably associated with β$_2$ in adult cardiomyocytes. **b** Schematic of a scenario where distinct Ca$_V$β isoforms preferentially associate with particular α$_1$-subunit types to mediate different functions (i). Knockdown of one β-isoform may lead to β reshuffling that lessens the functional impact of elimination of the particular Ca$_V$β subunit (ii). By contrast, targeted posttranslational inhibition of the channel complex based on the β isoform would yield a qualitatively different result that more accurately reflects the functional logic of Ca$_V$β molecular diversity in the cell (iii).

Ca$_V$β$_2$, Ca$_V$β$_3$, or Ca$_V$β$_4$ (Fig. 2c, d). To gain insights into the basis for the apparent selectivity of nb.E8 for β$_1$, we probed for the structural determinants on Ca$_V$β$_1$ necessary for nb.E8 binding. Ca$_V$βs share a common architecture in which conserved SH3 and NK domains are separated by a variable HOOK region, and flanked by variable N and C-termini (V1 and V2, respectively)[31–33] (Fig. 2e). We first divided Ca$_V$β$_1$ into two modules− NT-SH3-HOOK-SH3[β5] (with the essential SH3 β5 strand following the HOOK domain) and NK-CT− and tagged both moieties with Cerulean. Flow-FRET revealed that nb.E8 bound NT-SH3-HOOK-SH3[β5] but not NK-CT (Fig. 2f). Ca$_V$βs differ in their N-termini, with Ca$_V$β$_1$ having an extended N-terminus, raising expectations that this region might hold the essential determinants for selective nb.E8 binding. However, a construct with the Ca$_V$β$_1$ N-terminus deleted, SH3-HOOK-SH3[β5] retained robust binding to nb.E8, ruling out an important role of this region for the association (Fig. 1f). We reasoned that the SH3 domain was more likely to bind nb.E8 than the disordered HOOK region and thus searched for sequence variations among Ca$_V$β subunits in the conserved SH3 domain. We identified a region on β$_{1b}$ SH3 domain encompassing a 3$_{10}$ helix (η$_1$) that contains several unique residues compared to other β isoforms (Fig. 2g). A chimeric construct, replacing the 3$_{10}$ helix of Ca$_V$β$_3$ with the equivalent regions from β$_1$ (i.e., Ca$_V$β$_3$[η$_{β1}$]) was sufficient to reconstitute strong binding to nb.E8 (Fig. 2h, i). Conversely, the reverse chimera, Ca$_V$β$_{1b}$[η$_{β3}$], in which the η$_1$ region of β$_{1b}$ was replaced with that of β$_3$ displayed a strongly reduced binding affinity to nb.E8 (Fig. 2h). The residual binding of Ca$_V$β$_{1b}$[η$_{β3}$] to nb.E8 suggested other determinants contribute to the high binding affinity of the Ca$_V$β$_1$/nb.E8 interaction. We turned to X-ray crystallography to gain more in-depth structural insights into how

nb.E8 selectively binds Ca$_V$β$_1$, and how this contrasts with nb.F3 which non-selectively binds all four Ca$_V$β subunits.

## Structural basis for nb.F3 and nb.E8 association with Ca$_V$β subunits

To understand the differential nanobody recognition modes, we crystalized Ca$_V$β complexes containing the non-selective nanobody nb.F3 and the Ca$_V$β$_{1b}$-selective nanobody nb.E8 with two target Ca$_V$βs, Ca$_V$β$_{2a}$[34] and Ca$_V$β$_{1b}$, and determined their structures using X-ray crystallography. Ca$_V$β$_{2a}$ was crystallized as a 1:1 complex with nb.F3 (Supplementary Fig. 1a), whereas Ca$_V$β$_{1b}$ was crystallized as a 1:1:1 complex with nb.F3 and nb.E8 (Supplementary Fig. 1b). Both complexes diffracted X-rays to 2.0 Å resolution and were solved by molecular replacement (Supplementary Fig. 1c, d; Supplementary Table 1). The structures show that both nanobodies primarily target the Ca$_V$β SH3 domain but do so by binding to opposite sides of the structure (Fig. 3a, b).

In the nb.F3:Ca$_V$β$_{2a}$ complex, nb.F3 displays the canonical nanobody architecture (Supplementary Fig. 1e, f) and Ca$_V$β$_{2a}$ is largely unchanged from its AID-bound form[35] with the exception of a displacement of the SH3 α2 helix and the resolution of the 263–283 loop spanning the η2 and α4 NK domain helices and ten residues of the C-terminal V3 domain (Root mean squared deviation Cα, RMSD$_{Cα}$ 0.568 Å) (Supplementary Fig. 1g). The nb.F3:Ca$_V$β$_{2a}$ complex showed that nb.F3 engages Ca$_V$β$_{2a}$ using two interaction sites encompassing ~forty nb.F3 residues (Fig. 3a; Supplementary Fig. 2a, b). Site 1 comprises interactions with the three complementarity determining regions (CDRs) CDR1, CDR2

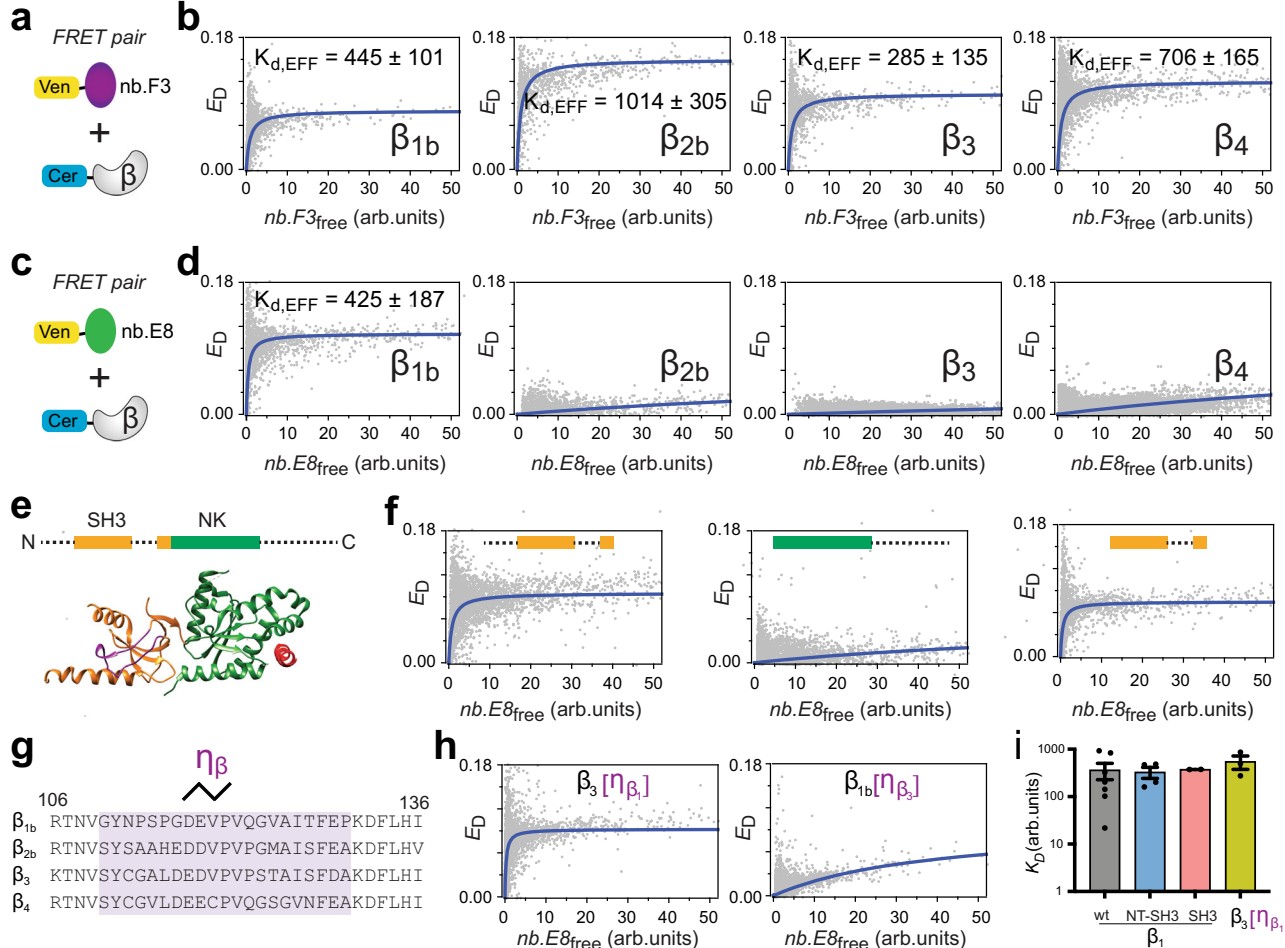

**Fig. 2 | Identification of nb.E8 as a $Ca_V\beta_1$-selective nanobody. a** Schematic of fluorescence resonance energy transfer (FRET) pair, Venus-nb.F3 + Cerulean-$Ca_V\beta$. **b** Flow cytometry FRET (flow-FRET) scatter plots with FRET efficiency ($E_D$) plotted vs free nb.F3 concentration. The blue lines fit a 1:1 binding model to the data. Data show nb.F3 binds all four $Ca_V\beta$ subunit isoforms. $K_{d,EFF}$, effective dissociation constant. **c** Schematic of Venus-nb.E8 + Cerulean-$Ca_V\beta$. **d** Flow-FRET scatter plots and binding curves show nb.E8 exclusively binds $Ca_V\beta_1$. **e** *Top*, schematic of $Ca_V\beta$ structural organization. Conserved src homology 3 (SH3, orange) and nucleotide kinase (NK, green) domains are bounded by variable N- and C-termini, and separated by a HOOK domain. Bottom, cryoEM structure of $Ca_V\beta_{1a}$ (PDB: 5GJV). **f** Flow-FRET scatter plots and binding curves for Cerulean-nb.E8 co-expressed with Venus-tagged $\beta_{1b}$[NT-SH3-HOOK] (left), $\beta_{1b}$[NK-CT] (middle), and $\beta_{1b}$[SH3-HOOK] (right). **g** Alignment of region within SH3 domains of human $Ca_V\beta$ subunits. Highlighted region is a 21-residue sequence ($\eta_1$) that shows some variability amongst $Ca_V\beta$ isoforms. **h** Flow-FRET scatter plots and binding curves for Cerulean-nb.E8 co-expressed with Venus-tagged chimeras $\beta_3[\eta\beta_{1b}]$ (left) and $\beta_{1b}[\eta\beta_3]$ (right). **i** $K_{d,EFF}$ values. Data are means ± SEM. $n = 17{,}409$ cells over 7 independent experiments (grey bar), $n = 8881$ cells over 4 independent experiments (blue bar), $n = 15{,}099$ cells over 2 independent experiments (pink bar), and $n = 6487$ cells over 3 independent experiments (yellow bar). Source data are provided as a Source Data file.

and CDR3, the majority of which are made between nb.F3 CDR3 with Phe62, Lys98, Asn 101, and Trp104 from β1, β2, and β3 of the SH3 core and CDR2 with Tyr200, Asp221, and Val222 from the SH3 β5 strand contributed from the NK domain (Fig. 3c; Supplementary Fig. 2a). Site 1 buries 866 Å² and contains a mixture of hydrogen bonding and van der Waals interactions (Supplementary Fig. 2a). Remarkably, the CDR3 Tyr102-Trp103 segment binds a small SH3 domain surface pocket made by Phe62, Trp104, and Val222 previously identified as a potential protein–protein interaction site by virtue of crystallographically observed binding of the short $Ca_V\beta_{2a}$ V2/HOOK domain sequence, Arg207-Phe210 (RMPF) to this site[31,36]. Structural comparison shows that CDR3 Tyr102-Trp103 and the RMPF peptide make remarkably similar interactions even though their peptide chains run in opposite directions (Supplementary Fig. 3a). This observation highlights the potential of this site as a protein-protein interaction surface. In contrast to Site1, Site 2 is smaller (513.1Å²) and rather than CDR interactions, comprises interactions of nb.F3 framework strands C″, D, and E residues with the 275-295 loop

located between the η2 and α4 NK domain elements. Notably, this loop is disordered in other $Ca_V\beta_{2a}$ structures[31,35] (Supplementary Fig. 1f).

Nb.F3 also uses Sites 1 and 2 to bind $Ca_V\beta_{1b}$, and the way nb.F3 binds to $Ca_V\beta_{1b}$ in the nb.F3:nb.E8:$Ca_V\beta_{1b}$ complex is very similar to nb.F3:$Ca_V\beta_{2a}$ components (RMSD$_{C\alpha}$ = 0.731 Å) (Supplementary Fig. 3b). Overall, the conformations and interactions of Site 1 are conserved and bury a similar amount of surface area (869 Å²). CDR3 makes the same interactions as in the nb.F3:$Ca_V\beta_{2a}$ complex (Supplementary Fig. 3c), while there are small differences in the interactions with CDR2 (Supplementary Fig. 2b). Although Site 2 in the $Ca_V\beta_{1b}$ complex remains primarily a site involving framework residues, there are differences from nb.F3:$Ca_V\beta_{2a}$. Most notably, there is an absence of interactions with the equivalent of the $Ca_V\beta_{2a}$ 275–295 loop; as a consequence, Site 2 buries much less surface area (146 Å²). Most of residues in Site 1 and Site 2 are highly conserved amongst different $Ca_V\beta$s, explaining the relative lack of specificity for nb.F3.

The nb.F3:nb.E8:$Ca_V\beta_{1b}$ complex shows that the β1 selective nanobody nb.E8, which has a longer CDR3 than nb.F3, also has the

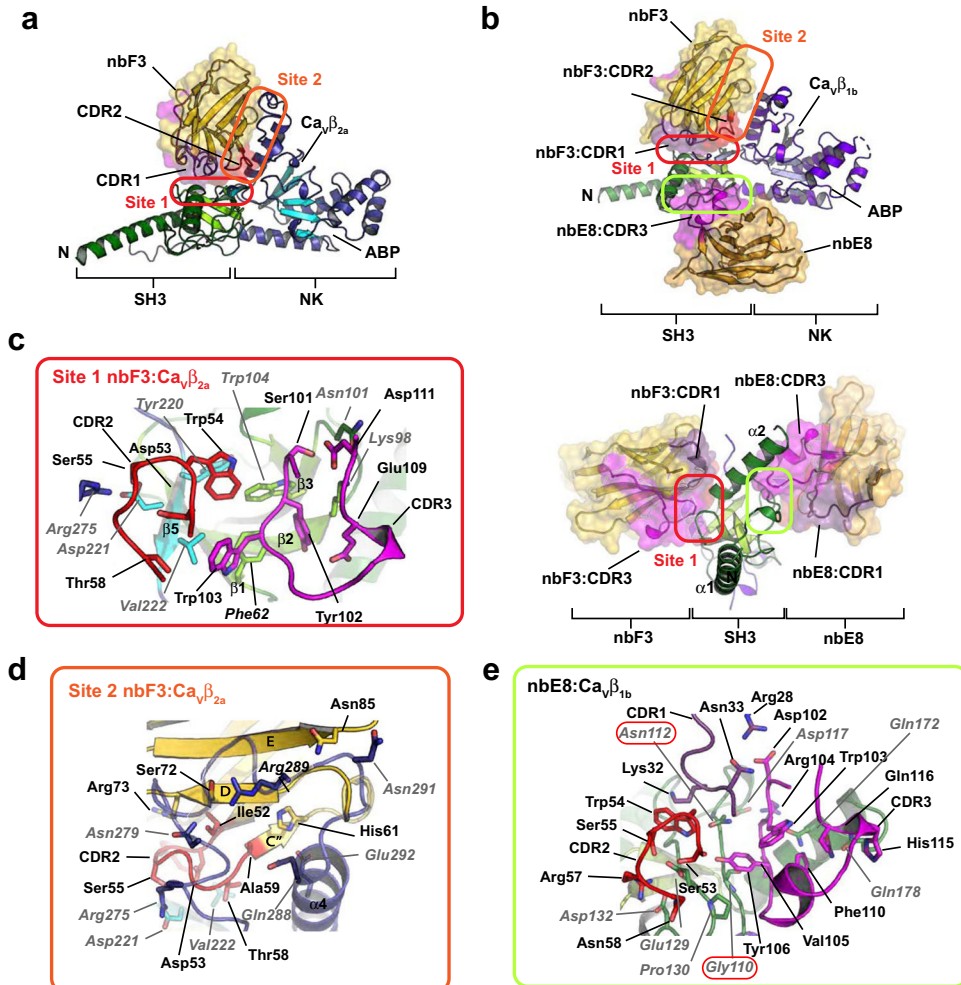

**Fig. 3 | Crystal structures of nb.F3 and nb.E8 in complex with $Ca_V\beta_1$.** **a** Cartoon diagram of the nb.F3:$Ca_V\beta_{2a}$ complex. $Ca_V\beta_{2a}$ SH3 (helices (dark green) and β-strands (light green)) and NK (helices (dark blue) and β-strands (cyan)) domains are indicated. Nb.F3 (yellow) is shown with a semi-transparent surface. Nb.F3 Site 1 and Site 2 binding regions are indicated by the red and orange ovals, respectively. **b** Cartoon diagram of nb.F3:nb.E8:$Ca_V\beta_{1b}$ complex. SH3 (helices (green) and β-strands (lime)) and NK (helices (purple) and β-strands (violet)) domains are indicated. Nb.F3 (yellow) and nb.E8 (yellow orange) are shown with a semi-transparent surface. Site 1 and Site 2 binding regions are indicated by the red and orange ovals,

respectively. Light green oval denotes nb.E8 binding site. Lower panel shows view from the SH3 domain N-terminus as indicated by the arrows. ABP indicates AID binding pocket location in (**a**) and (**b**), (**c**) View of nb.F3:$Ca_V\beta_{2a}$ Site 1 interactions. **d** View of nb.F3:$Ca_V\beta_{2a}$ Site 2 interactions. **e** View of nb.E8:$Ca_V\beta_{1b}$ interactions. Red ovals indicate $Ca_V\beta$ sites that differ among isoforms. Complementarity determining regions (CDRs) are colored similarly in all panels: CDR1 (purple), CDR2 (red), CDR3 (magenta). In (**c**–**e**) nanobody residues are in black, $Ca_V\beta$ residues are labeled in grey italics.

canonical nanobody structure (Supplementary Fig. 3e) and uses its three CDRs to bind to the $Ca_V\beta_{1b}$ SH3 domain face formed by the long loop between SH3 domain strands β1 and β2 opposite to the nb.F3 epitope (Fig. 3b). This interaction buries 853Å² and involves a mixture of hydrogen bonds, salt bridges, and van der Waals interactions (Supplementary Fig. 2c). One notable interaction is made by the Val103-Arg104 portion of CDR3. This loop wedges between the SH3 domain α2 helix and η1 loop allowing Arg104 to make a salt bridge with Asp117 and a series of backbone mediated hydrogen bonds to Asn112 and Gln172 of $Ca_V\beta_{1b}$, matching the region identified by our chimera studies (Fig. 2g, h), and thereby defining the positions that are important for the selectivity of nbE8 for $Ca_V\beta_1$ over other isoforms. These include residues in the center of the binding interface, Gly110, and Asn112 (Fig. 3e), and variations at Gly116 where other $Ca_V\beta$ isoforms have larger residues that would likely interfere with the nb.E8 Arg104 interactions (Fig. 2g; Supplementary Fig. 4). Hence, together the structural data define modes for both non-selective (nb.F3) and selective (nb.E8) recognition of $Ca_V\beta$ that exploit different aspects of the SH3 domain structure.

## Nb.E8 selectively inhibits recombinant $Ca_V1/Ca_V2$ channels reconstituted with $Ca_V\beta_1$

We evaluated whether nb.E8 (expressed in a P2A-CFP plasmid vector) affected HVACC functional expression by transient co-expression with recombinant human $Ca_V2.2$ ($\alpha_{1B} + \beta + \alpha_2\delta-1$) channels reconstituted in HEK293 cells (Fig. 4). Contemporaneous control experiments used HEK293 cells co-expressing CFP and recombinant $Ca_V2.2$ channels. We used path-clamp electrophysiology to record whole-cell currents through reconstituted $Ca_V2.2$ channels using 5 mM $Ba^{2+}$ as charge carrier. Control $Ca_V2.2$ channels reconstituted with any of the four $Ca_V\beta$ isoforms yielded robust whole-cell currents that activated at a threshold between −15 and −10 mV, and peaked at +10 mV (Fig. 4). $Ca_V2.2$ channels reconstituted with $\beta_1$ and nb.E8 yielded several differences in channel properties compared to control. First, the whole-cell current amplitude was strongly decreased ($I_{peak} = -102.8 \pm 15.78$ pA/pF, $n = 20$ for CFP; and $I_{peak} = -22.42 \pm 4.321$ pA/pF, $n = 16$ for nb.E8, $P = 6.5 \times 10^{-5}$, unpaired two-tailed Student's $t$ test) (Fig. 4a, b). Second, the voltage-dependence of channel activation

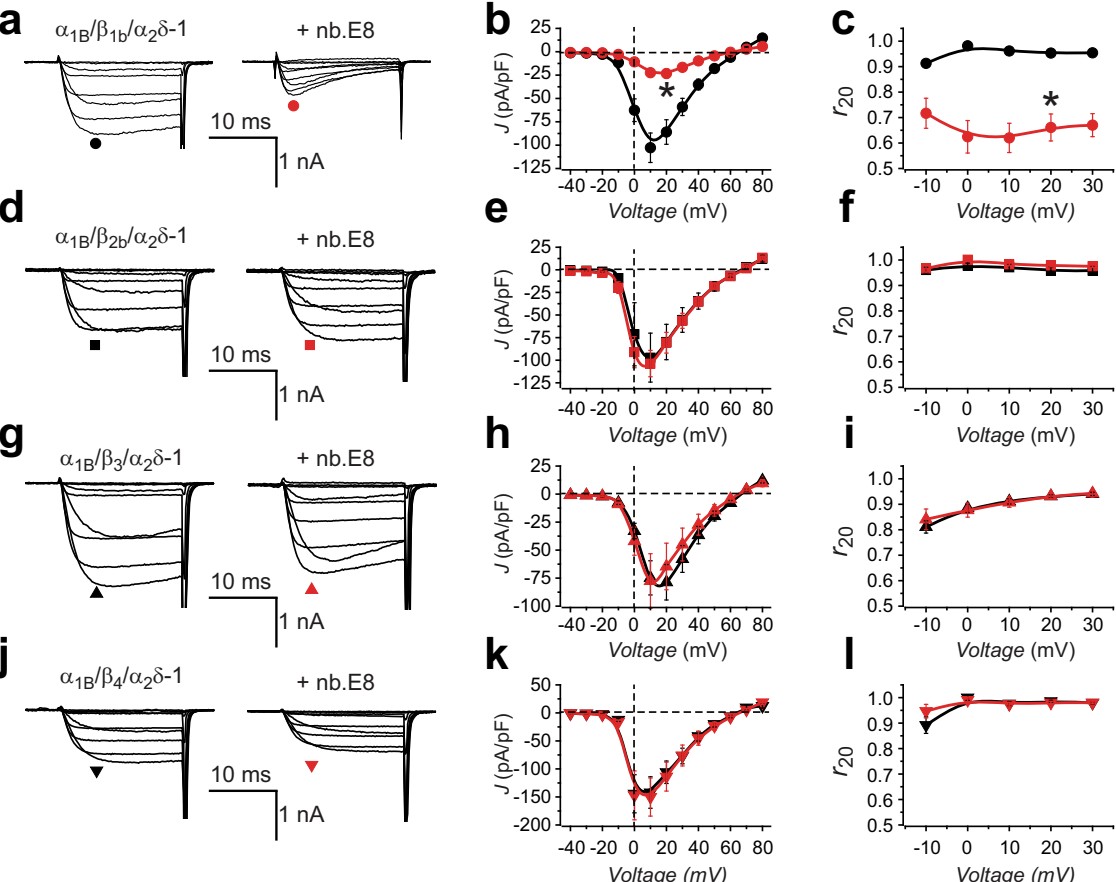

**Fig. 4 | Nb.E8 inhibits current amplitude and speeds inactivation in $Ca_V\beta_1$-associated $Ca_V2.2$ channels. a** Exemplar family of whole cell currents from HEK293 cells expressing reconstituted $Ca_V2.2$ channels ($\alpha_{1B}+\beta_{1b}+\alpha_2\delta$-1) in the absence (left) or presence (right) of nb.E8. **b** Population current density vs voltage (J-V) relationship for cells expressing $\alpha_{1B}+\beta_{1b}+\alpha_2\delta$-1 in the absence (black circles; $n = 20$ cells examined over 4 independent experiments) or presence (red circles; $n = 16$ cells examined over 4 independent experiments) of nb.E8. *$P = 6.5 \times 10^{-5}$, two-tailed unpaired $t$-test. **c** Fractional current remaining after 20 ms ($r_{20}$) at various test pulse voltages for cells expressing $\alpha_{1B}+\beta_{1b}+\alpha_2\delta$-1 in the absence (black circles; $n = 20$ cells examined over 4 independent experiments) or presence (red circles; $n = 13$ cells examined over 4 independent experiments) of nb.E8. *$P = 7.4 \times 10^{-5}$, two-tailed unpaired $t$ test. **d–f** Data for $\alpha_{1B}+\beta_{2b}+\alpha_2\delta$-1 ± nb.E8, same format as (**a–c**). **e** $n = 8$ cells examined over 3 independent experiments (black symbols) and $n = 6$ cells

examined over 3 independent experiments (red symbols). **f** $n = 7$ cells examined over 3 independent experiments (black symbols) and $n = 5$ cells examined over 3 independent experiments (red symbols). **g–i** Data for $\alpha_{1B}+\beta_3+\alpha_2\delta$-1 ± nb.E8, same format as **a–c**. **h** $n = 17$ cells examined over 4 independent experiments (black symbols) and $n = 7$ cells examined over 3 independent experiments (red symbols). (**i**) $n = 12$ cells examined over 4 independent experiments (black symbols) and $n = 6$ cells examined over 3 independent experiments (red symbols). **j–l** Data for $\alpha_{1B}+\beta_4+\alpha_2\delta$-1 ± nb.E8, same format as **a–c**. **k** $n = 7$ cells examined over 3 independent experiments (black symbols) and $n = 8$ cells examined over 3 independent experiments (red symbols). **l** $n = 12$ cells examined over 3 independent experiments (black symbols) and $n = 10$ cells examined over 3 independent experiments (red symbols). Data are means ± SEM. Source data are provided as a Source Data file.

was right-shifted as indicated by a + 10-mV shift in the $V_{0.5}$ of the tail activation curve (Supplementary Fig. 5). Third, channels expressed with nb.E8 displayed a faster rate of voltage-dependent inactivation (VDI) that was evident in exemplar traces (Fig. 4a), and quantified in population data as a decrease in the fractional current remaining after 20 ms ($r_{20} = I_{20}/I_{peak}$) across a range of test pulse voltages ($r_{20}$ at +10 mV = 0.96 ± 0.08, $n = 20$ for CFP; and $r_{20}$ at +10 mV = 0.62 ± 0.05, $n = 13$ for nb.E8, $P = 7.4 \times 10^{-5}$, unpaired two-tailed Student's $t$ test) (Fig. 4c). In sharp contrast, in $Ca_V2.2$ channels reconstituted with $\beta_2$, $\beta_3$, or $\beta_4$, co-expressing nb.E8-P2A-CFP produced no change in current amplitude or gating behavior compared to control (Fig. 4d–l).

We investigated whether nb.E8 could selectively inhibit other $Ca_V1/Ca_V2$ channel isoforms as long as they were reconstituted with $\beta_1$ subunit. Indeed, we found that nb.E8 significantly inhibited $Ca_V2.1$, $Ca_V2.3$, and $Ca_V1.3$ channels that were reconstituted with $\beta_{1b}$ but not $\beta_{2b}$, $\beta_3$, or $\beta_4$ subunits. Thus, the ability of nb.E8 to selectively inhibit $\beta_1$-bound channels is a general feature for $Ca_V1/Ca_V2$ channels.

## Nb.E8 inhibits surface density and $P_o$ of $Ca_V2.2$ channels reconstituted with $Ca_V\beta_1$

The whole-cell current ($I$) is related to microscopic channel properties by the relation $I = N \times i \times P_o$, where $N$ is the total number of channels at the cell surface, $i$ is the unitary current amplitude, and $P_o$ is the single-channel open probability. We first determined whether a decrease in $N$ contributes to the impact of nb.E8 on $I$. To address this, we applied a flow cytometry-based assay to measure surface density of $Ca_V2.2$ channels; the method utilizes an engineered $\alpha_{1B}$ that harbors a tandem repeat of the high-affinity $\alpha$-bungarotoxin-binding site (BBS) (derived from the nicotinic acetylcholine receptor) in the extracellular domain IV S5-S6 loop[6,7]. We co-expressed BBS-$\alpha_{1B}$ with $\beta_{1b}$-YFP and $\alpha_2\delta$-1 to simultaneously measure surface (Alexa-647 conjugated $\alpha$-bungarotoxin) and total $\beta_{1b}$ (YFP fluorescence) levels in non-permeabilized HEK293 cells using flow cytometry (Fig. 5a, b; Supplementary Fig. 6). Compared to control cells in which CFP was co-expressed with BBS-$\alpha_{1B}$ + $\beta_{1b}$-YFP + $\alpha_2\delta$-1, expression of nb.E8 significantly decreased $Ca_V2.2$ surface density (50% reduction) while only moderately affecting $\beta_{1b}$ expression (-20% reduction in YFP fluorescence) (Fig. 5b, c).

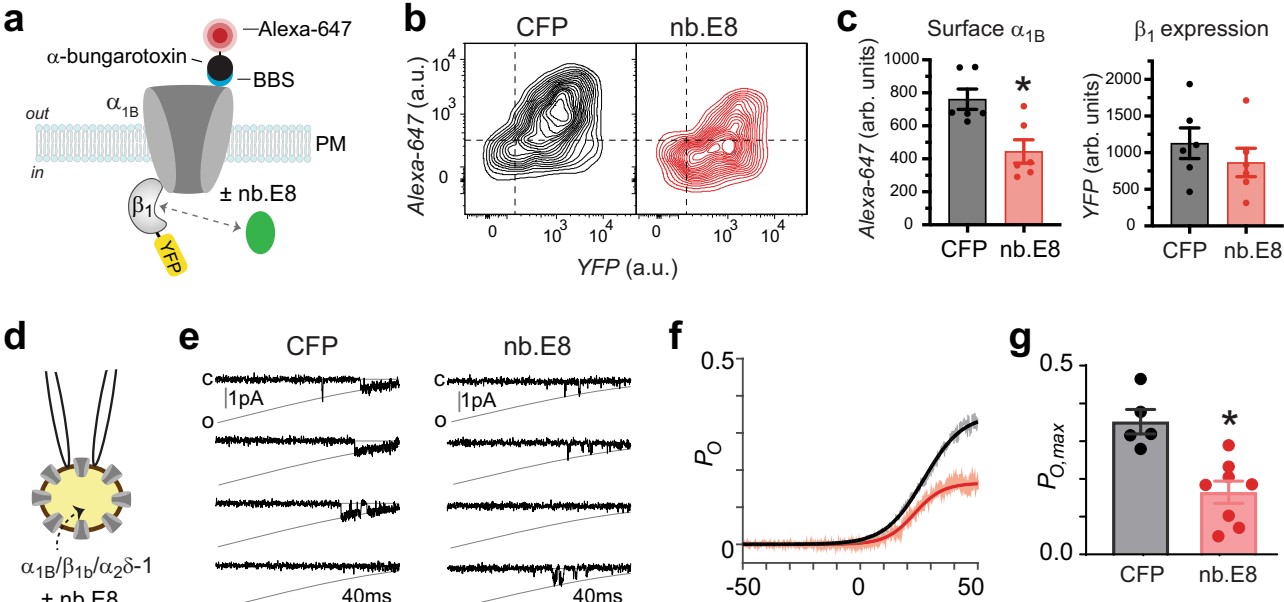

**Fig. 5 | Nb.E8 inhibits Ca$_V$2.2 surface density and single channel $P_o$. a** Schematic of BBS-$\alpha_{1B}$ + $\beta_{1b}$-YFP ± nb.E8. **b** Flow cytometry contour plots showing Ca$_V$2.2 surface density (Alexa647) vs $\beta_{1b}$ expression (YFP) in HEK293 cells expressing $\alpha_{1B}$+$\beta_{1b}$+$\alpha_2\delta$−1 with either CFP (left) or nb.E8 (right). **c** Bar charts showing impact of nb.E8 on Ca$_V$2.2 surface density ($n = 6$ over 3 independent experiments for both CFP and nb.E8 groups) and $\beta_{1b}$-YFP expression ($n = 6$ over 3 independent experiments for both CFP and nb.E8 groups). *$P = 0.0052$ compared to CFP control, two-tailed unpaired $t$ test. **d** Schematic of cell-attached single channel recording.

**e** Exemplar single channel recordings evoked by slow ramp protocols in HEK293 cells expressing $\alpha_{1B}$+$\beta_{1b}$+$\alpha_2\delta$−1 with either CFP (left) or nb.E8 (right). **f** Ensemble average open probability vs voltage ($P_o$-$V$) relationships in cells expressing $\alpha_{1B}$+$\beta_{1b}$+$\alpha_2\delta$−1 with either CFP (gray) or nb.E8 (red). **g** Bar charts showing impact of nb.E8 on maximal single-channel open probability, $P_{o,max}$ ($n = 5$ over 3 independent experiments for CFP, and $n = 8$ over 3 independent experiments for nb.E8). *$P = 0.0017$ compared to CFP control, two-tailed unpaired $t$ test. Data are means ± SEM. Source data are provided as a Source Data file.

To determine whether changes in $i$ and/or $P_o$ contribute to nb.E8-mediated decrease in $I$ we turned to single-channel recordings which enable direct measurement of these parameters (Fig. 5d). We used 40 mM Ba$^{2+}$ as charge carrier in the cell-attached patch clamp mode and a slow-voltage ramp protocol to obtain stochastic channel openings, which reflect near-steady state $P_o$ at each voltage[12,37]. Control Ca$_V$2.2 channels ($\alpha_{1B}$ + $\beta_{1b}$ + $\alpha_2\delta$−1) that were expressed with CFP had robust openings with a maximal $P_o$ ($P_{o,max}$) of 0.35, while channels co-expressed with nb.E8 exhibited a $P_{o,max}$ that was approximately half of the control ($P_{o,max} = 0.35 ± 0.032$, $n = 5$ for CFP; $P_{o,max} = 0.16 ± 0.029$, $n = 8$ for nb.E8, $P = 0.0017$ unpaired two-tailed Student's $t$ test) (Fig. 5e–g). The unitary current amplitude was unchanged. Overall, these data indicate that nb.E8 selectively inhibits $\beta_{1b}$-bound Ca$_V$2.2 channels by decreasing both $N$ and $P_o$.

### Nb.E8-HECT$_{Nedd4L}$ (Chisel-1) selectively ablates $\beta_1$-associated HVACC currents

Though nb.E8 intrinsically acts as a potent inhibitor of $\beta_1$-associated HVACCs the elimination of current is incomplete (Fig. 4a, b). We previously showed that fusing the HECT domain of Nedd4L to nb.F3 created a construct, Ca$_V$-aβlator, that completely eliminated HVACCs irrespective of the β-subunit isoform with which they were reconstituted[28]. The mechanism involved Ca$_V$-aβlator targeting to auxiliary β subunits in channel complexes and producing a ubiquitination dependent intracellular relocation of the channel complex. We wondered whether similar engineering of nb.E8 would generate an equally potent, but $\beta_1$-selective, HVACC inhibitor. Accordingly, we created a chimeric nb.E8-HECT$_{Nedd4L}$ (Chisel-1) construct downstream of CFP-P2A to permit separate expression of reporter fluorescence and engineered nb.E8 proteins, respectively (Fig. 6a). When co-expressed with $\alpha_{1C}$ + $\beta_{1b}$, Chisel-1 decreased expression of both $\beta_{1b}$ and $\alpha_{1C}$, most likely due to ubiquitin-mediated degradation, as supported by the increased ubiquitination of both subunits (Fig. 6c–f). Whole-cell patch

clamp recordings revealed that Chisel-1 essentially eliminated all current through recombinant Ca$_V$2.2 channels reconstituted with $\beta_{1b}$ ($J_{peak} = -102.8 ± 15.78$ pA/pF, $n = 20$ for CFP; and $J_{peak} = -1.65 ± 0.34$ pA/pF, $n = 10$ for Chisel-1, $P = 3.82 × 10^{-6}$, unpaired two-tailed Student's $t$ test) (Fig. 6g, h). In sharp contrast, Chisel-1 had no effect on Ca$_V$2.2 channels reconstituted with $\beta_2$, $\beta_3$, or $\beta_4$ subunits (Fig. 6i–k). We obtained similar results for recombinant Ca$_V$2.1, Ca$_V$2.3, and Ca$_V$1.3 channels− Chisel-1 abolished current through these channels when they were reconstituted with $\beta_{1b}$, but was completely ineffective when the channels were formed with $\beta_2$, $\beta_3$, or $\beta_4$ (Fig. 7).

Flow cytometry surface staining assay showed that in cells co-expressing BBS-$\alpha_{1B}$ + $\beta_{1b}$-YFP Chisel-1 reduced the surface density of BBS-$\alpha_{1B}$ beyond that achieved with nb.E8 (~60% decrease), while only moderately inhibiting $\beta_{1b}$ expression (Supplementary Fig. 7).

### Probing Chisel-1 efficacy and selectivity in skeletal and cardiac muscle cells

We wondered whether Chisel-1 could potently and selectively inhibit endogenous $\beta_1$-bound native HVACCs in primary cells. Skeletal and cardiac muscle cells provided ideal systems to address this question due to certain unique characteristics. First, they both have intricate cytoarchitectures with complicated intracellular environments specialized for their contractile function. This complex cytoplasmic milieu thus provides a good challenge for monitoring the efficacy of Chisel-1. Second, adult skeletal muscle fibers exclusively express Ca$_V$1.1 ($\alpha_{1S}$) in complex with $\beta_{1a}$ (Fig. 8a), whereas adult ventricular cardiomyocytes predominantly express Ca$_V$1.2 ($\alpha_{1C}$) in complex with $\beta_2$ (Fig. 8e) with no contribution from $\beta_1$[26,38]. Thus, the efficacy of Chisel-1 can be determined by how effectively it inhibits Ca$_V$1.1 in skeletal muscle, while its selectivity can be deduced from how ineffective it is against Ca$_V$1.2 in ventricular cardiomyocytes.

We performed in vivo electroporation of flexor digitorum brevis (FDB) fibers of adult mice with plasmids encoding either CFP (control)

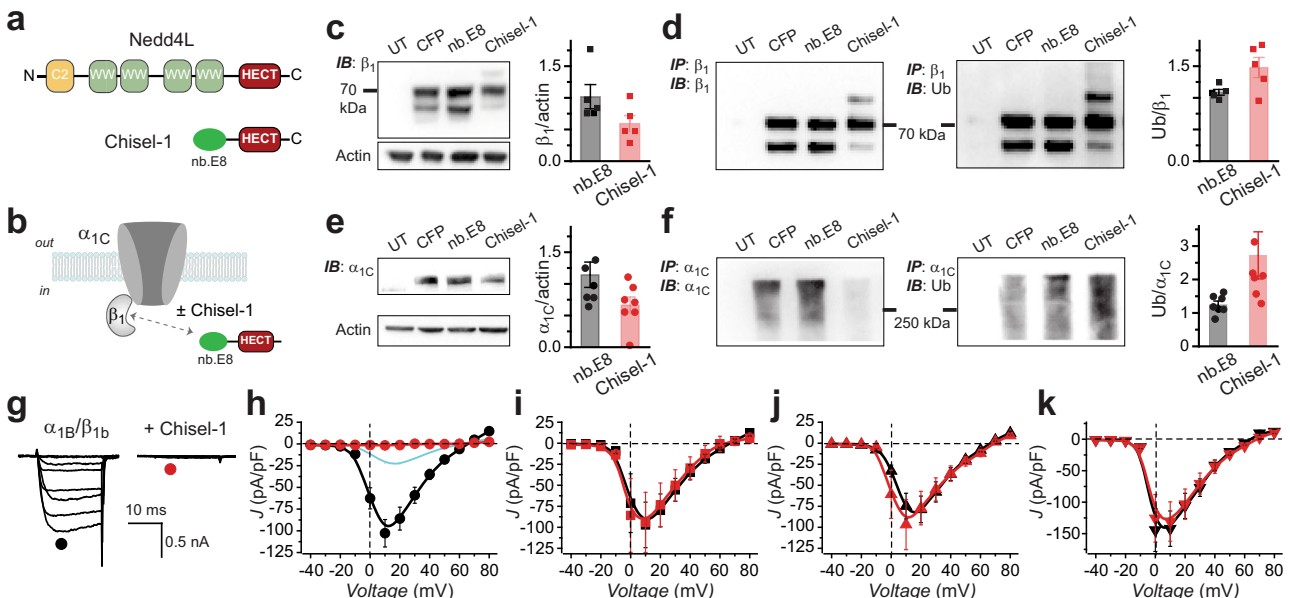

**Fig. 6 | Chisel-1 selectively eliminates recombinant $Ca_V2.2$ channels reconstituted with $Ca_V\beta_1$. a** Schematic showing modular structures of Nedd4L and Chisel-1. **b** Schematic of $\alpha_{1C} + \beta_{1b} \pm$ Chisel-1. **c** Left, $\beta_{1b}$ immunoblot in untransfected cells (UT), or cells transiently transfected with $\alpha_{1C} + \beta_{1b}$ and CFP, nb.E8 or Chisel-1. Right, Bar charts showing impact of Chisel-1 on normalized $\beta_{1b}$ expression ($n = 5$ independent experiments each for CFP, nb.E8, and Chisel-1 groups). **d** Left, $\beta_{1b}$ pulldown followed by $\beta_{1b}$ immunoblot. Middle, $\beta_{1b}$ pulldown followed by ubiquitin immunoblot. Right, Bar chart showing Chisel-1 enhances ubiquitination of $\beta_{1b}$ ($n = 5$ independent experiments each for CFP, nb.E8, and Chisel-1 groups). **e** Left, $\alpha_{1C}$ immunoblot. Right, impact of Chisel-1 on $\alpha_{1C}$ normalized expression ($n = 6$ independent experiments each for CFP, nb.E8, and Chisel-1 groups). **f** Left, $\alpha_{1C}$ pulldown followed by $\alpha_{1C}$ immunoblot. Middle, $\alpha_{1C}$ pulldown followed by ubiquitin immunoblot. Right, Bar chart showing Chisel-1 enhances ubiquitination of $\alpha_{1C}$ ($n = 6$ independent experiments each for CFP, nb.E8, and Chisel-1 groups). **g** Exemplar

family of whole-cell currents in HEK293 cells expressing $\alpha_{1B}+\beta_{1b}+\alpha_2\delta-1$ in the absence (left) or presence (right) of Chisel-1. **h** Population current density vs voltage (J-V) relationship for cells expressing $\alpha_{1B}+\beta_{1b}+\alpha_2\delta-1$ in the absence (black circles; $n = 20$ over 5 independent transfections) or presence (red circles; $n = 10$ over three independent transfections) of Chisel-1. **i** J-V relationship for $\alpha_{1B}+\beta_{2b}+\alpha_2\delta-1 \pm$ Chisel-1, same format as (**h**). (For CFP, $n = 8$ over 3 independent transfections, black symbols; for Chisel-1, $n = 6$ over 3 independent transfections, red symbols). **j** J-V relationship for $\alpha_{1B}+\beta_3+\alpha_2\delta-1 \pm$ Chisel-1, same format as (**h**) (For CFP, $n = 17$ over 3 independent transfections, black symbols; for Chisel-1, $n = 8$ over 3 independent transfections, red symbols). **k** J-V relationship for $\alpha_{1B}+\beta_4+\alpha_2\delta-1 \pm$ Chisel-1, same format as (**h**) (for CFP, $n = 7$ over 3 independent transfections, black symbols; for Chisel-1, $n = 10$ over 3 independent transfections, red symbols). Data are means ± SEM. Source data are provided as a Source Data file.

or Chisel-1 (expressed in a CFP-P2A cassette). Ten days after electroporation, muscle fibers were dissociated into individual cells many of which expressed CFP fluorescence, confirming transgene expression (Fig. 8b). Control skeletal muscle myocytes expressing CFP displayed large whole cell $Ca^{2+}$ currents that were completely eliminated in cells expressing Chisel-1 ($I_{peak} = -12.97 \pm 1.51$ pA/pF, $n = 13$ for CFP; $I_{peak} = -0.02 \pm 0.61$ pA/pF, $n = 13$ for Chisel-1, $P = 6.37 \times 10^{-7}$) (Fig. 8c). Gating current measurements revealed Chisel-1 significantly decreased, but did not completely eliminate, intramembrane charge movement, consistent with a reduced $Ca_V1.1$ surface density compared to control cells expressing CFP ($Q_{max} = 30.97 \pm 1.75$ pA/pF, $n = 7$ for CFP; $Q_{max} = 15.47 \pm 2.17$ pA/pF, $n = 8$ for Chisel-1, $P = 9.32 \times 10^{-5}$) (Fig. 8d). Chisel-1 also reduced stimulus-evoked rhod2 $Ca^{2+}$ transients by 50% compared to control skeletal muscle fibers expressing CFP (Supplementary Fig. 8), consistent with excitation-contraction coupling in skeletal muscle being mediated by voltage-induced $Ca^{+2}$ release rather than the $Ca^{2+}$-induced $Ca^{2+}$ release found in heart cardiomyocytes[39]. These results demonstrate that Chisel-1 eliminates $Ca^{2+}$ current through $Ca_V\beta_1$- associated $Ca_V1.1$ channels in skeletal muscle and this effect is partially mediated through a reduction in the channel surface density. The discrepancy between the near complete ablation of whole-cell current and the 50% reduction in $Ca_V1.1$ surface density suggests that Chisel-1 inhibits the $P_o$ of channels remaining at the surface even more strongly than we found for nb.E8. This could be due to the enhanced ubiquitination which has been shown to inhibit gating of some channels[40].

In sharp contrast with the elimination of $Ca_V1.1$ current in skeletal muscle, adenovirus-mediated expression of Chisel-1 in adult guinea pig

ventricular cardiomyocytes had no effect on whole-cell $Ca_V1.2$ functional expression ($I_{peak} = -7.08 \pm 0.97$ pA/pF, $n = 15$ for mCherry; $I_{peak} = -8.39 \pm 1.33$ pA/pF, $n = 8$ for Chisel-1) (Fig. 8f, g). Importantly, we previously showed that $Ca_V$-aβlator (comprised of nb.F3 fused to Nedd4L HECT domain) eliminates $Ca_V1.2$ current in adult guinea pig ventricular cardiomyocytes[28]. $Ca_V1.2$ in adult cardiac myocytes can target to the cell surface and support functional currents independent of $Ca_V\beta$ binding[26]. Thus, it was possible that Chisel-1 could interfere in some way with $Ca_V\beta$ functional interaction with $Ca_V1.2$ in cardiomyocytes that would, nevertheless, not register as a frank change in current amplitude. However, interaction of $Ca_V\beta$ with $Ca_V1.2$ is obligatory for up-regulation of cardiac $I_{Ca,L}$ by activated PKA[27], offering a path to test whether Chisel-1 disrupted $\alpha_{1C}/\beta$ interaction in heart cells. We found no difference in the magnitude of forskolin-induced increase in $I_{Ca,L}$ between control cardiomyocytes expressing mCherry ($4.19 \pm 0.32$ fold) and cells expressing Chisel-1 ($4.12 \pm 0.66$ fold) (Fig. 8h) indicating the $\alpha_{1C}/\beta_{2b}$ interaction remained intact. Overall, the sharp dichotomy of Chisel-1 effects on $I_{CaL}$ in skeletal versus cardiac myocytes demonstrates the efficacy of this unique tool in selectively eliminating current through $Ca_V\beta_1$-bound channels in native cells.

## Chisel-1 inhibits excitation-transcription coupling in hippocampal neurons

Hippocampal neurons express multiple $Ca_V$ channel $\alpha_1$ (including $Ca_V1.2$, $Ca_V1.3$, $Ca_V2.1$, $Ca_V2.2$, and $Ca_V2.3$) and $Ca_V\beta$ ($Ca_V\beta_1$-$Ca_V\beta_4$) subunits[41–45], representing a context in which there is the bewildering possibility of at least 20 molecularly distinct $Ca_V$ channel $\alpha_1/\beta$ subunit combinations. Chisels that are able to selectively inhibit $Ca_V$ channels

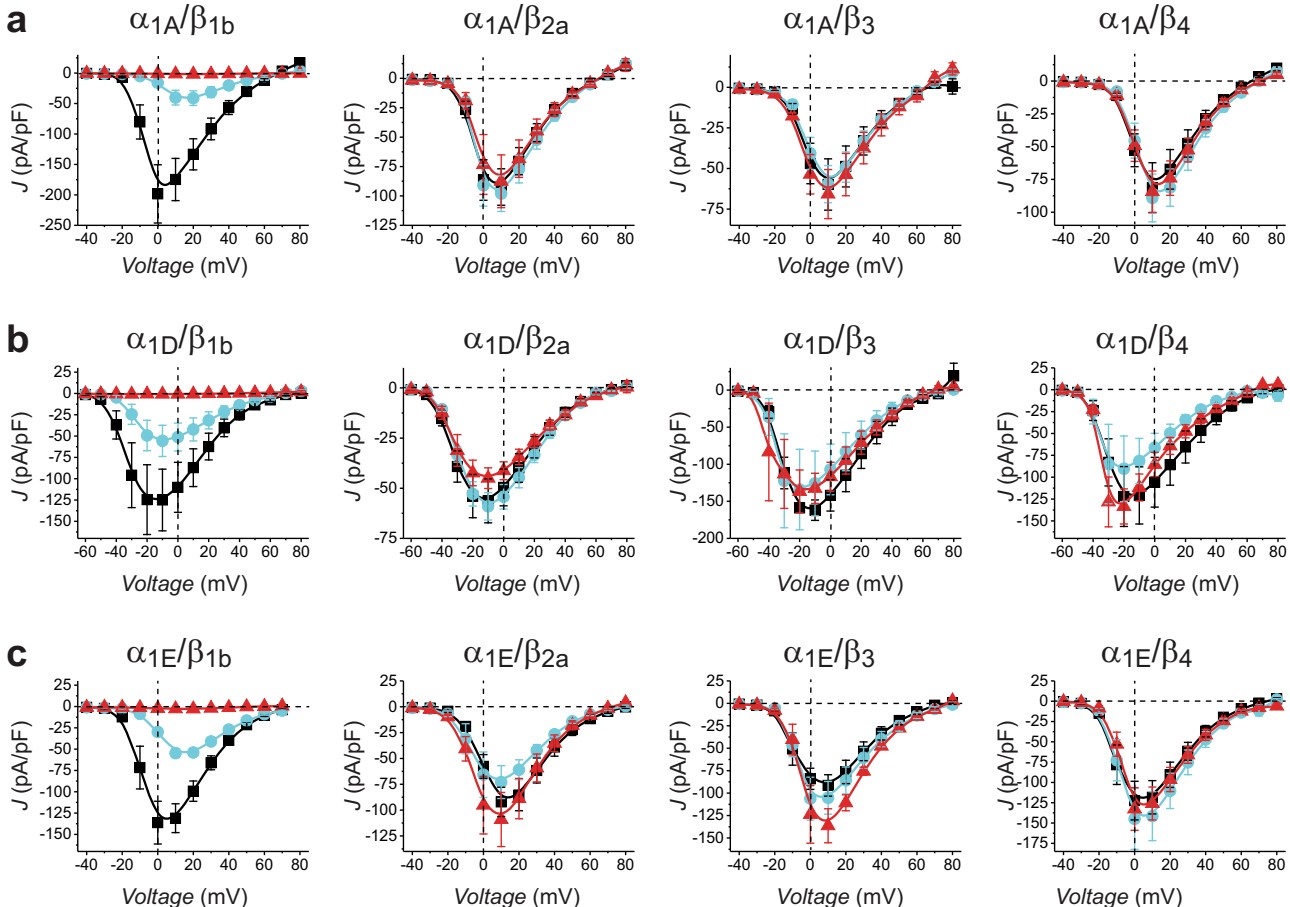

**Fig. 7 | Generality of nb.E8 and Chisel-1 selective inhibition of recombinant Ca_V1/Ca_V2 channels reconstituted with Ca_Vβ_1.** **a** Population current density vs voltage (J-V) relationship for cells expressing $\alpha_{1A}$+β+$\alpha_2\delta$−1 and co-expressed with CFP (black symbols), nb.E8 (cyan symbols) or Chisel-1 (red symbols): $\alpha_{1A}$+$\beta_{1b}$+$\alpha_2\delta$−1 (CFP, $n$ = 6; nb.E8, $n$ = 6, Chisel-1, $n$ = 3); $\alpha_{1A}$+$\beta_{2b}$+$\alpha_2\delta$−1 (CFP, $n$ = 7; nb.E8, $n$ = 6, Chisel-1, $n$ = 5); $\alpha_{1A}$+$\beta_3$+$\alpha_2\delta$−1 (CFP, $n$ = 11; nb.E8, $n$ = 6, Chisel-1, $n$ = 6); $\alpha_{1A}$+$\beta_4$+$\alpha_2\delta$−1 (CFP, $n$ = 10; nb.E8, $n$ = 5, Chisel-1, $n$ = 8). **b** J-V relationship for cells expressing $\alpha_{1D}$+β +$\alpha_2\delta$−1 with CFP (black symbols), nb.E8 (cyan symbols) or Chisel-1 (red symbols):

$\alpha_{1D}$+$\beta_{1b}$+$\alpha_2\delta$−1 (CFP, $n$ = 8; nb.E8, $n$ = 9, Chisel-1, $n$ = 5); $\alpha_{1D}$+$\beta_{2b}$+$\alpha_2\delta$−1 (CFP, $n$ = 8; nb.E8, $n$ = 8, Chisel-1, $n$ = 7); $\alpha_{1D}$+$\beta_3$+$\alpha_2\delta$−1 (CFP, $n$ = 3; nb.E8, $n$ = 5, Chisel-1, $n$ = 5); $\alpha_{1D}$+$\beta_4$+$\alpha_2\delta$−1 (CFP, $n$ = 7; nb.E8, $n$ = 10, Chisel-1, $n$ = 4). **c** J-V relationship for cells expressing $\alpha_{1E}$+β+$\alpha_2\delta$−1 with CFP (black symbols), nb.E8 (cyan symbols) or Chisel-1 (red symbols): $\alpha_{1E}$+$\beta_{1b}$+$\alpha_2\delta$−1 (CFP, $n$ = 6; nb.E8, $n$ = 8, Chisel-1, $n$ = 6); $\alpha_{1E}$+$\beta_{2b}$+$\alpha_2\delta$ −1 (CFP, $n$ = 7; nb.E8, $n$ = 5, Chisel-1, $n$ = 5); $\alpha_{1E}$+$\beta_3$+$\alpha_2\delta$−1 (CFP, $n$ = 8; nb.E8, $n$ = 7, Chisel-1, $n$ = 4); $\alpha_{1E}$+$\beta_4$+$\alpha_2\delta$−1 (CFP, $n$ = 12; nb.E8, $n$ = 5, Chisel-1, $n$ = 6). Data are means ± SEM. Source data are provided as a Source Data file.

based on the identity of their associated Ca_Vβ isoform could be an exceptional tool to help discriminate the functional logic of Ca_Vβ molecular diversity in neurons. Accordingly, we determined whether Chisel-1 would be effective in revealing physiological effects mediated by Ca_Vβ_1-bound Ca_V channels in cultured hippocampal neurons.

Lentiviral-mediated expression of mCherry-P2A-Chisel-1 resulted in a substantial reduction in Ca_Vβ_1 immunofluorescence as compared to control neurons expressing mCherry alone (Fig. 9a). Western blot confirmed the diminished expression of Ca_Vβ_1 in neurons expressing Chisel-1 (Fig. 9b). By contrast, Chisel-1 had no impact on expression levels of Ca_Vβ_2 or Ca_Vβ_3. However, neurons expressing Chisel-1 showed a higher expression of Ca_Vβ_4 compared to control (Fig. 9b). Interestingly, β_4-null lethargic mice selectively displayed a compensatory increase in β_1b expression[46], suggesting a specific reciprocal regulation of these two auxiliary Ca_Vβ isoforms.

Hippocampal neurons expressing Chisel-1 displayed a diminished KCl-evoked $Ca^{2+}$ transient amplitude consistent with a decrease in whole-cell current (Fig. 9c, d). We examined whether Ca_Vβ_1-bound channels participate in excitation-transcription coupling in hippocampal neurons[47]. In neurons pre-treated with NBQX + AP5 + TTX, a 3-minute exposure to 40 mM KCl resulted in a 6-fold increase in phosphoCREB immunoreactivity in the nucleus, consistent with previous reports (Fig. 9e, f). The 40 KCl-induced increase in nuclear

phosphoCREB was strongly reduced in the presence of a cocktail of Ca_V1/Ca_V2 channel blockers (nisoldipine + ω-conotoxin GVIA + ω-aga-toxin IVA) fitting with the known role of VGCCs in this phenomenon (Fig. 9f). Ca_V-αβlator inhibited excitation-induced phosphoCREB signal in the nucleus of hippocampal neurons to a similar extent as the cocktail of Ca_V channel blockers (Fig. 9f). With these controls established, we discovered that hippocampal neurons expressing Chisel-1 also displayed a robust reduction in the 40 KCl-evoked increase in nuclear phosphoCREB staining, indicating that Ca_Vβ_1-bound channels play a prominent role in excitation-transcription coupling in hippocampal neurons (Fig. 9f).

## Discussion

This work describes the development and application of Chisel-1, a unique tool that potently and selectively inhibits Ca_Vβ_1-bound HVACC in cells by promoting targeted ubiquitination of the channel complex, decreasing channel surface density ($N$), and reducing single-channel $P_o$. The mechanism of action of Chisel-1 is qualitatively different from conventional gene knockout or shRNA knockdown approaches and yields complementary information to these established methods. More broadly, our success with Chisel-1 supplies a blueprint for developing Chisels that inhibit not only HVACCs based on the Ca_Vβ isoform (β_2 − β_4) they are associated with, but also the myriad other

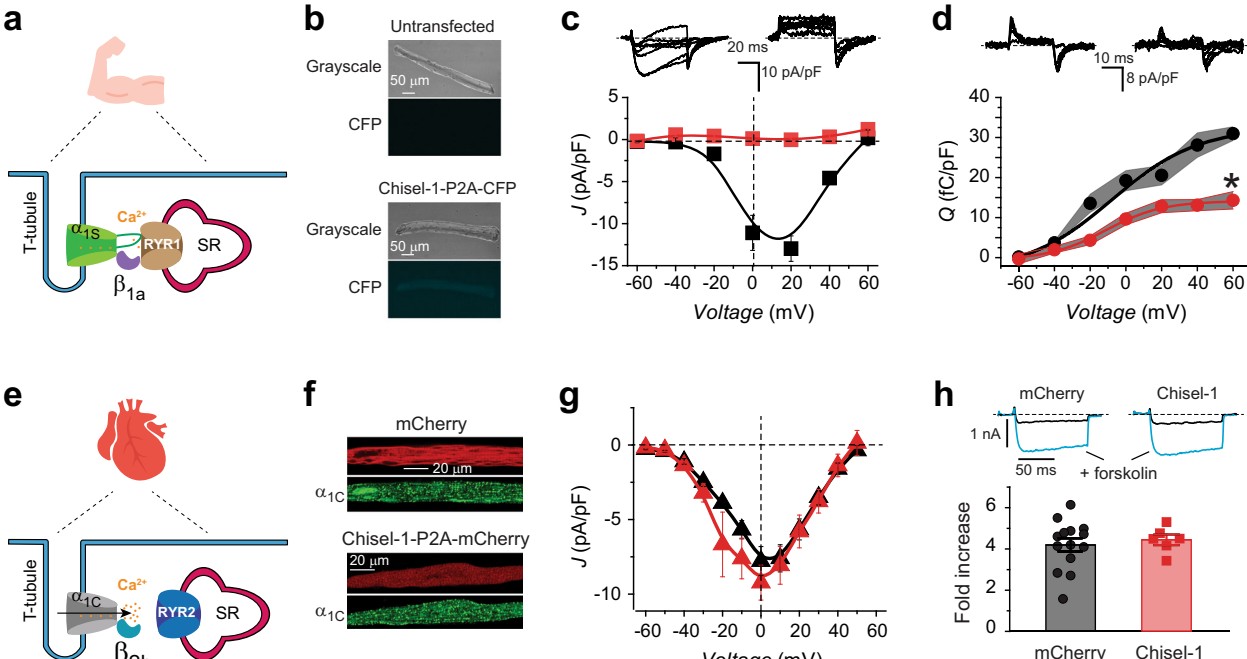

**Fig. 8 | Chisel-1 selectively eliminates current through endogenous $Ca_V\beta_1$-associated HVACCs. a** Schematic of skeletal muscle Cav1.1 complex. **b** Images of isolated flexor digitorum brevis (FDB) fibers either untransfected (top) or transfected with Chisel-1-P2A-CFP (bottom). **c** Top, exemplar whole-cell currents from isolated FDB fibers expressing CFP (left) or Chisel-1 (right). Bottom, Population J-V curves from isolated FDB fibers expressing CFP (black squares; $n = 13$ over 3 independent experiments) or Chisel-1 (red squares; $n = 13$ over 3 independent experiments). **d** Top, exemplar gating currents from isolated FDB fibers expressing CFP (left) or Chisel-1 (right). Bottom, Population $Q$-$V$ curves from isolated FDB fibers expressing CFP (black circles; $n = 7$ over 2 independent experiments) or Chisel-1 (red circles; $n = 8$ over 2 independent experiments). $*P = 9.32 \times 10^{-5}$ compared to

CFP control, two-tailed unpaired $t$ test. **e** Schematic of ventricular cardiomyocyte $Ca_V1.2$ complex. **f** Confocal images of cardiomyocytes expressing mCherry (*top*) or Chisel-1-P2A-mCherry (bottom). **g** Population J-V curves from isolated ventricular myocytes expressing mCherry (black triangles; $n = 13$ over 3 independent experiments) or Chisel-1 (red triangles; $n = 11$ over 3 independent experiments). **h** Top, exemplar whole-cell currents from ventricular myocytes expressing mCherry (left) or Chisel-1 (right) before (black) and after (cyan) application of 1 μM forskolin. Bottom, lack of effect of Chisel-1 on forskolin induced increase in $I_{Ca,L}$ in ventricular myocytes (mCherry, $n = 14$; Chisel-1, $n = 5$). Data are means ± SEM. Source data are provided as a Source Data file.

multi-subunit ion channel complexes. Overall, this method significantly adds to the arsenal of tools available to probe physiological/pathophysiological roles of ion channel subunit molecular diversity, and to develop novel genetically-encoded ion channel inhibitors as research tools or potential therapeutics.

$Ca^{2+}$ influx through HVACCs controls a rich variety of physiological responses in excitable cells. The diversified responses ensuing from a singular signaling event, $Ca^{2+}$ influx, is mediated in part by molecular heterogeneity of HVACC pore-forming $\alpha_1$ and auxiliary $Ca_V\beta$ subunits. Selective small molecule or toxin blockers available for the different $Ca_V\alpha_1$ isoforms have been invaluable in delineating their distinctive physiological roles[2]. By contrast, there are no equivalent molecules or methods that can post-translationally eliminate HVACC function based on the identity of the associated $Ca_V\beta$ isoform. $Ca_V\beta_1$ - $Ca_V\beta_4$ subunits have some overlapping functions such as facilitating surface membrane targeting of co-expressed $\alpha_1$ subunits, shifting the voltage-dependence of channel activation in the hyperpolarizing direction, and increasing channel $P_o$[3]. However, the individual β-subunit isoforms also confer distinctive rates of inactivation and steady-state inactivation properties to HVACCs that hint at unique functional roles in native cells[3]. Results from knockout mice and model organisms confirm the notion of unique physiological roles for different $Ca_V\beta$ isoforms. For example, $Ca_V\beta_1$ knockout mice die at birth due to asphyxia, a consequence of $\beta_{1a}$ being the sole $Ca_V\beta$ isoform expressed in skeletal muscle[21]. β1-null zebrafish (*relaxed*) are paralyzed owing to loss of skeletal muscle EC coupling[48]. Swapping in a different $Ca_V\beta$ isoform for $Ca_V\beta_{1a}$ in either dysgenic mouse or β1-null zebrafish skeletal muscle restored whole cell Cav1.1 current but did not recover tetrad formation

or EC coupling[49], explicitly demonstrating the principle of both overlapping and unique functions of distinct $Ca_V\beta$ isoforms. Similarly, embryonic knockout of $\beta_2$ is lethal due to impaired development of the heart and an absence of cardiac contractions[23]. Transgenic expression of $\beta_2$ in the hearts of $\beta_2$-null mice rescued viable animals which were, nevertheless, deaf[24] and experienced visual impairments reminiscent of patients with congenital stationary night blindness[22]. The interpretation of functional effects of $Ca_V\beta$ knockouts in cell types such as neurons that express multiple $Ca_V\alpha_1$ and $Ca_V\beta$ isoforms can be more complicated. For example, $Ca_V\beta_3$ knockout mice displayed alterations in learning and memory tasks, yet showed no changes in whole-cell $Ca^{2+}$ currents from hippocampal neurons[50]; and selectively exhibited a dampened response to pain initiated by chemical inflammation, but with no overt changes in HVA $Ca^{2+}$ currents[25]. These conflicting results from $Ca_V\beta_3$ knockout studies likely reflect the phenomenon of $Ca_V\beta$ subunit reshuffling in which other $Ca_V\beta$ isoforms present in neurons bind to $Ca_V\alpha_1$ subunits and occupy slots that would otherwise have been engaged by $Ca_V\beta_3$. Post-translational inhibition of HVACCs based on their specific constituent $Ca_V\beta$ isoform as embodied in the Chisel concept would be expected to limit the confounding effects of $Ca_V\beta$ reshuffling and reveal the full scope of the functional logic of $Ca_V\beta$ molecular diversity. Consistent with this notion, Chisel-1 reduced the amplitude of high KCl-evoked $Ca^{2+}$ transients and strongly inhibited excitation-transcription coupling in hippocampal neurons. These results provide an impetus to develop Chisels that are selective for $Ca_V\beta_2$, $Ca_V\beta_3$, and $Ca_V\beta_4$ isoforms as a focus for future studies. Beyond their utility as tools to decipher physiological roles of $Ca_V\beta$ isoforms, Chisels have potential therapeutic applications as genetically-encoded

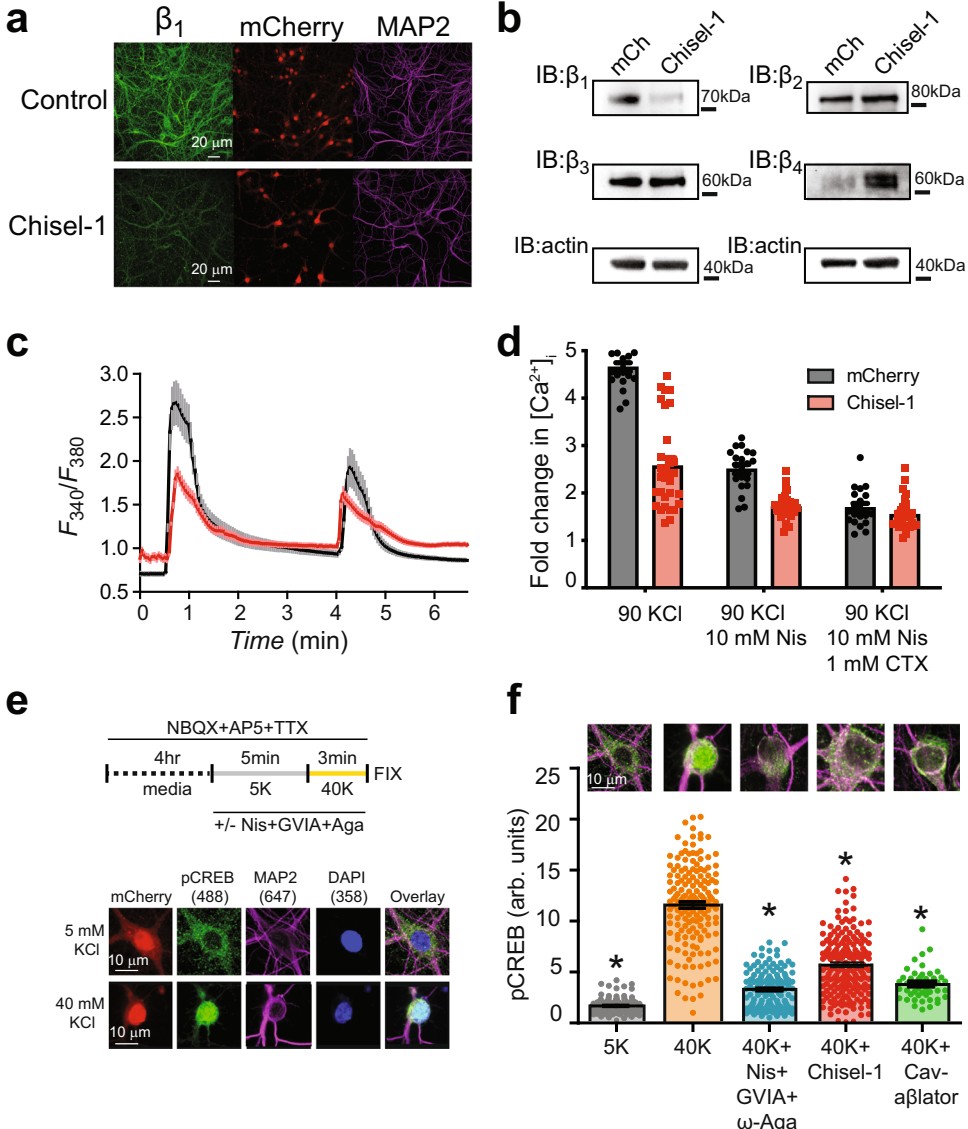

**Fig. 9 | Chisel-1 inhibits Ca²⁺ influx and excitation-transcription coupling in hippocampal neurons. a** Representative hippocampal neuron cultures expressing mCherry (top row) or Chisel-1-P2A-mCherry (bottom row) with immuno-fluorescence detection of $\beta_{1b}$ (green) and MAP2 (purple). Images are representative of three independent experiments. **b** Representative Ca$_V\beta$ isoform immunoblots from hippocampal neurons expressing either mCherry or Chisel-1. Immunoblots are representative of three independent experiments. **c** Time course of mean 90 KCl-evoked Ca²⁺ transients in hippocampal neurons expressing either mCherry (black line) or Chisel-1 (red trace). **d** Bar charts showing impact of Chisel-1 on 90 KCl-evoked Ca²⁺ transients before and after the addition of Ca$_V$1/Ca$_V$2 channel blockers (mCherry, $n = 20$ over 3 independent experiments; Chisel-1, $n = 30$ over 3 independent experiments). **e** Top, schematic of experimental protocol for assessing excitation-transcription coupling in cultured hippocampal neurons. Bottom,

exemplar images showing phosphoCREB (pCREB), microtubule associated protein 2 (MAP2), and DAPI staining of fixed mCherry-expressing hippocampal neurons exposed to either 5 or 40 mM KCl. NBQX, 2,3-dioxo-6-nitro-7-sulfamoyl-benzo[f] quinoxaline; TTX tetrodotoxin, AP5 2-Amino-5-phosphonopentanoic acid. **f** Bar chart showing 40 KCl-induced nuclear translocation of pCREB signal and its inhibition by a cocktail of Ca$_V$1/Ca$_V$2 channel inhibitors and Chisel-1: 5 K, $n = 146$; 40 K, $n = 166$; 40 K + nisoldipine + ω-conotoxin GVIA + ω-agatoxin, $n = 139$; Chisel-1, $n = 189$; Ca$_V$-aβlator, $n = 48$. * significantly different compared to 40 mM KCl condition by one-way ANOVA and Tukey's multiple comparison test: 5 K, $P = 0$; 40 K + nisoldipine + ω-conotoxin GVIA + ω-agatoxin, $P = 2.1 \times 10^{-8}$; Chisel-1, $P = 5.5 \times 10^{-9}$; Ca$_V$-aβlator, $P = 0$. Data are means ± SEM. Source data are provided as a Source Data file.

HVACC inhibitors with selectivity for distinct Ca$_V\beta$ isoforms. As an example of their potential advantages, we consider Ca$_V\beta_3$ knockout mice which display reduced anxiety but increased aggression[51]. There are distinctive neuronal circuits for the two types of behaviors suggesting the possibility it may be possible to reduce anxiety with localized expression of a Chisel-3 without inducing aggressive behaviors.

Our results are also noteworthy for providing the first proof of concept of using an unmodified Ca$_V\beta$-targeted nanobody to inhibit current amplitude and modulate gating properties of HVACCs. Nb.E8 intrinsically inhibited current amplitude, sped up inactivation, and

right-shifted the voltage-dependence of activation of Ca$_V\beta_1$-bound channels. Understanding the principles of how nb.E8 alters the biophysical properties of HVACCs may enable the rational design of small molecules that can achieve similar objectives. Interestingly, the binding site for nb.E8 is on Ca$_V\beta_1$ SH3 rather than the NK domain that mediates high affinity binding to $\alpha_1$ subunits[31–33]. Some previous efforts to develop Ca$_V$1/Ca$_V$2 channel inhibitors have focused on disrupting the association of Ca$_V\beta$ NK with the AID in $\alpha_1$ subunits using either small molecules[52,53] or peptides[35]. Our findings suggest the possibility that small molecules that target Ca$_V\beta$ SH3 domains could be a novel

class of effective $Ca_V1/Ca_V2$ channel inhibitors. Missense mutations in HVACC $Ca_V\alpha_1$ subunits that causes devastating neurological and cardiovascular diseases often do so by altering channel gating properties[54–57]. Thus, molecules that can correct mutation-induced gating changes in HVACCs have prospective use as therapeutics for *CACNA1-* and *CACNA2-*associated rare ion channelopathies. Our results also motivate the search for other $Ca_V\beta$-targeted nanobodies that can modulate HVACC channel gating in unique ways.

The identification of a nanobody that modulates HVACCs in a $Ca_V\beta$-isoform-specific manner gives reason to wonder whether there are endogenous proteins in cells that mediate similar functions, as supported by observation of a commonly targeted protein-protein interaction on the SH3 domain surface (Supplementary Fig. 3a)[36]. The principle of natural proteins that powerfully modulate function of HVACCs via binding to auxiliary $Ca_V\beta$ subunits is already well-established with RGK (Rad, Rem, Rem2, Gem) monomeric G-proteins which act as potent HVACC inhibitors[6,58,59]. Indeed, relief of this inhibition by PKA-mediated phosphorylation is the mechanism underlying β-adrenergic stimulation of cardiac contractility essential for the fight-or-flight response[60]. Beyond HVACCs, other ion channel families are also often comprised of pore-forming $\alpha_1$ proteins assembled with auxiliary subunits that have multiple isoforms. Thus, the principle of post-translational inhibition of channels in an auxiliary subunit-isoform-specific manner is one that is generally applicable. Our description of Chisel-1 provides a blueprint for how this critical gap in deducing the functions of multi-subunit membrane proteins can be bridged.

## Methods

### Nanobody generation
$Ca_V\beta_1$ and $Ca_V\beta_3$ purifications and nanobody generation were described previously[61]. Briefly one llama was immunized with an initial injection of 600 μg purified $Ca_V\beta_{1b}$ and $Ca_V\beta_3$, with four boosters of 200 μg each protein administered every other week (Capralogics Inc., Hardwick, MA). 87 days after the first immunization, lymphocytes were isolated from blood sample and nanobody sequences amplified using a two-step nested PCR. The amplified Vhh genes were cloned into the phagemid plasmid, pComb3xSS (gift from Carlos Barbas; Addgene plasmid # 63890)[62], and a phage display library constructed using electrocompetent TG1 *E. coli* cells (Lucigen). Three rounds of phage display were performed[63] using 100 nM biotinylated $Ca_V\beta_{1b}$ as bait on neutravidin-coated Nunc-Immuno plates (Thermo Scientific). Multiple clones of interest including nb.F3 and nb.E8 were cloned into mammalian expression systems for further characterization.

### Molecular biology and plasmid construction
For flow-FRET assays, candidate $Ca_V\beta$ nanobodies were cloned into pCDNA3.1 and upstream of a Venus marker using EcoRI/HindIII cloning sites. $Ca_V\beta$ subunits were cloned into the PiggyBac CMV mammalian expression vector and downstream of a Cerulean marker using NotI/MluI cloning sites. Subsequent $Ca_V\beta$ cloning (SH3, NK modules, chimeras) were done using Gibson cloning[64].

A customized bicistronic vector (xx-P2A-CFP) was synthesized in pUC57 vector in which coding sequence for P2A peptide was sandwiched between an upstream multiple cloning site and enhanced cyan fluorescent protein (CFP) (Genewiz). The xx-P2A-CFP fragment was amplified by PCR and cloned into the PiggyBac CMV mammalian expression vector (System Biosciences) using NheI/NotI sites. To generate nb.E8 -P2A-CFP, we PCR amplified the coding sequence for nb.E8 and cloned it into xx-P2A-CFP using NheI/AflII sites. A similar backbone was created in the PiggyBac CMV mammalian expression vector in which CFP-P2A-xx contained a multiple cloning site downstream of the P2A site (Genewiz). Nb.E8 was PCR amplified and ligated into the vector with BglII/AscI sites. The HECT domain of human

Nedd4L[65] (a gift from Joan Massague, Addgene plasmid # 27000) consisting of residues 596-975 was PCR amplified and inserted downstream of nb.E8 using AscI/AgeI sites. Mutagenesis of C942S was accomplished using site-directed mutagenesis.

$\alpha_{1B}$-BBS, harboring two tandem 13 residue bungarotoxin-binding sites (SWRYYESSLEPYPD) in the domain IV S5-S6 extracellular loop, was a kind gift from Dr. Steven Ikeda (NIAAA). $\alpha_{1C}$ and $\alpha_{1C}$-BBS, and $\alpha_{1C}$-BBS-YFP have been described previously[6,66].

### Virus generation
Generation of nb.E8-IRES-mCherry and nb.E8-Nedd4L-IRES-mCherry adenoviruses was performed by Vector Biolabs (Malvern, PA).

For lentivirus construction, we followed established protocols[67] using packaging plasmids kindly provided by Dr. David Baltimore. CalFectin was used to transfect the packaging plasmids VSV-G and ΔP along with the insert containing plasmid nb.E8-p2a-mCherry. Confluent HEK293T cells grown in 10 cm dishes were transfected and the media was exchanged the following morning to hippocampal culture media containing: Neurobasal media (Thermo Fisher Scientific), B-27 supplement (Thermo Fisher Scientific), and Glutamax supplement (Thermo Fisher Scientific). Media was collected approximately 48 h after transfection and centrifuged at 500 g for five minutes to pellet debris. Supernatants were collected, aliquoted, and stored at −80 °C.

### HEK293 cell culture and transfection
Human embryonic kidney (HEK293) cells were a kind gift from the laboratory of Dr. Robert Kass (Columbia University). Low passage HEK293 cells were cultured at 37 °C in DMEM supplemented with 5% fetal bovine serum (FBS) and 100 mg/mL of penicillin−streptomycin. HEK293 cell transfection was accomplished using the calcium phosphate precipitation method. Briefly, plasmid DNA was mixed with 7.75 μL of 2 M $CaCl_2$ and sterile deionized water (to a final volume of 62 μL). The mixture was added dropwise, with constant tapping to 62 μL of 2x HEPES buffered saline containing (in mM): HEPES 50, NaCl 280, $Na_2HPO_4$ 1.5, pH 7.09. The resulting DNA−calcium phosphate mixture was incubated for 20 min at room temperature and then added dropwise to HEK293 cells (60−80% confluent). Cells were washed with $Ca^{2+}$-free phosphate buffered saline after 4−6 h and maintained in supplemented DMEM.

### Guinea pig cardiomyocyte isolation and culture
Isolation of adult guinea pig cardiomyocytes was performed in accordance with the guidelines of Columbia University Animal Care and Use Committee. Prior to isolation, plating dishes were precoated with 15 μg/mL laminin (Gibco). Adult male Hartley guinea pigs (Charles River) were euthanized with 5% isoflurane, hearts were excised and ventricular myocytes isolated by first perfusing in KH solution (mM): 118 NaCl, 4.8 KCl, 1 $CaCl_2$ 25 HEPES, 1.25 $K_2HPO_4$, 1.25 $MgSO_4$, 11 glucose, .02 EGTA, pH 7.4, followed by KH solution without calcium using a Langendorff perfusion apparatus. Enzymatic digestion with 0.3 mg/mL Collagenase Type 4 (Worthington) with 0.08 mg/mL protease and .05% BSA was performed in KH buffer without calcium for six minutes. After digestion, 40 mL of a high $K^+$ solution was perfused through the heart (mM): 120 potassium glutamate, 25 KCl, 10 HEPES, 1 $MgCl_2$, and .02 EGTA, pH 7.4. Cells were subsequently dispersed in high $K^+$ solution. Healthy rod-shaped myocytes were cultured in Medium 199 (Life Technologies) supplemented with (mM): 10 HEPES (Gibco), 1x MEM non-essential amino acids (Gibco), 2 L-glutamine (Gibco), 20 D-glucose (Sigma Aldrich), 1% vol $vol^{-1}$ penicillin-streptomycin-glutamine (Fisher Scientific), 0.02 mg/mL Vitamin B-12 (Sigma Aldrich) and 5% (vol/vol) FBS (Life Technologies) to promote attachment to dishes. After 5 h, the culture medium was switched to Medium 199 with 1% (vol/vol) serum, but otherwise supplemented as described above. Cultures were maintained in humidified incubators at 37 °C and 5% $CO_2$.

## In vivo gene transfer via muscle electroporation

For in vivo electroporation experiments, animal procedures and protocols were reviewed and approved by the Institutional Animal Care and Use Committees of the University of Maryland. Male C57BL/6 J mice (Charles River, Wilmington, MA) were used. All mice used (9 mice) were between 30–60 days of age. Environmental conditions were maintained with a 12-h light/dark cycle and constant temperature (21–23 °C) and humidity (55 ± 10%). The cages contained corncob bedding (Harlan Teklad 7902) and environmental enrichment (cotton nestlet). Mice were supplied with dry chow (irradiated rodent diet; Harlan Teklad 2981) and water ad libitum.

Electroporation was carried out on 4-week-old C57BL mice. The intramuscular injection of various DNA plasmids was conducted, with minor modifications, according to previous reports[68,69]. Briefly, one footpad of an anesthetized mouse is injected subcutaneously with 20–30 μl of 3 mg/ml hyaluronidase through a 33-gauge needle. Then, 1 to 2 h later, the mouse is again anesthetized and ~40 μg of plasmid DNA is injected into the footpad. Ten minutes later, two surgical stainless-steel electrodes are placed subcutaneously close to the proximal and distal tendons of the flexor digitorum brevis (FDB) muscle and 20 pulses of 100 V/cm, 20 ms in duration, are applied at 1 Hz via a commercial high current capacity output stage (ECM 830, BTX, Harvard Apparatus, Holliston, MA). One to two weeks later, single muscle fibers are enzymatically dissociated from the injected FDB muscles and cultured as described below.

## Skeletal muscle fiber culture

Culture of flexor digitorum brevis (FDB) was carried out as previously described[70,71]. Animals were euthanized by asphyxiation via CO2 followed by cervical dislocation according to protocols approved by the University of Maryland Institutional Animal Care and Use Committee. Briefly, the FDB muscle was isolated from male adult mice, enzymatically dissociated with collagenase type I (Sigma-Aldrich, St. Louis, MO) in MEM (Life Technologies, Carlsbad, CA) with 10% FBS, and 50 μg/ml gentamicin for 3–4 h at 37 °C. Muscle was then gently triturated to separate fibers in MEM with FBS and gentamicin. Fibers were plated in MEM culture media with 10% FBS on glass-bottomed dishes (Matek Cor. Ashland, MA, Cat. No. P35G-1.0-14-C,) coated with laminin (Thermo Fisher, Rockford, IL, Cat. No. 23017-015). Fibers were maintained in culture for 1 to 2 days at 37 °C, 5% CO2 prior to the experiments. Positively transfected fibers were identified by the CFP expression profile.

## Two-electrode voltage clamp (TEVC)

The TEVC was used to measure non-linear capacitive currents and L-type Ca2+ current elicited by step depolarizations. Muscle fibers (<500 μm in length) were chosen and visualized on a Zeiss Axiovert 200 M inverted microscope. The external recording solution composition to measure non-linear capacitive currents was (in mM): 150 TEA-CH3SO3, 10 HEPES, 0.5 CaCl2, 1 MgCl2, 0.5 CdCl2 and 0.5 CoCl2, 0.001 TTX, 0.5 4-aminopiridine, 0.025 BTS (N-benzyl-p-toluene sulphonamide; Sigma-Aldrich, St Louis, MO, Cat No. 203895), pH adjusted to 7.4 with CsOH. To measure Ca2+ currents, 10 mM Ca2+ was used as charge carrier and Cd2+ and Co2+ were not added to the recording solution. The current injecting electrode (V1) was filled with 2 M cesium aspartate and voltage measuring electrode (V2) was filled with 3 M cesium chloride as previously described[69]. Microelectrode V1 was placed at the middle of the selected fiber, and V2 was positioned halfway between the middle and the end of the selected fiber.

We used an AxoClamp 900 A and Axon Digidata 1550B low-noise digitizer (Molecular Devices, San Jose, CA, USA), HS-9A x1 (V1) and HS-9A x0.1 (V2) headstages and borosilicate glass (Warner Instruments, Cat No. G150TF-3) with resistances of 10–20 MΩ when filled with the electrode solution. Once the fibers were impaled with both microelectrodes, cells were held at −80 mV. Fibers with signs of clamp error,

such as unstable holding current or rapid drifts on holding potential, were rejected from the analysis. Measurements started 3 min after TEVC clamp configuration was established. Voltage protocols were generated and current responses were digitized and stored using Clampex and Clampfit (version 11, Molecular Devices, San Jose, CA, USA). Command pulses were delivered at 30 s intervals to the levels and duration indicated in each figure from a holding potential of −80 mV, unless otherwise indicated. Currents were typically low-pass-filtered at 3–10 kHz (3-pole Bessel filter). Currents were sampled at 10 kHz. Linear capacitive and ionic currents were routinely subtracted by a P/4 protocol[72]. Gating charge moved during each test depolarization ($Q_{ON}$) was quantified by calculating the area under the curve of each trace of non-linear current using the post-transient level of each trace as a steady-state value of non-linear ionic current. Total charge moved during repolarization ($Q_{OFF}$) was calculated similarly[73,74]. Total charge movement was normalized to the linear fiber capacitance, which was determined by measuring linear capacitive current elicited by a ± 5 mV test pulse from the holding potential and integrating the area under the capacitive current trace to estimate Q.

Data analysis was performed using Clampfit 8.0 (Molecular Devices, San Jose, CA, USA). Further data evaluation, non-linear fitting and statistical analysis were conducted using OriginPro 2020b software. The I–V plots from muscle fibers were fitted to a Boltzmann-Ohmic function, described by the following equation[75]:

$$I(V) = G_{max}(V - V_{rev}) / \left[1 + \exp\left(-\left(\frac{V - V_{half}}{k}\right)\right)\right] \quad (1)$$

where $G_{max}$ is the maximum conductance, $V$ is the membrane potential, $V_{rev}$ is the reversal potential, $V_{half}$ is the half-activation potential, and $k$ is a measure of the steepness.

Similarly, the Q-V relationship of each individual fiber was fitted to a single Boltzmann function, as described by the equation:

$$Q(V) = Q_{max} / (1 + \exp((V_{half} - V)/k)) \quad (2)$$

where $Q_{max}$ gives the maximum charge movement, $V_{half}$ defines the potential where Q = 0.5 of $Q_{max}$ and $1/k$ is a measure of the steepness of the Q–V relationship.

## Hippocampal isolation and culture

E18 Sprague Dawley Rat hippocampal tissue was purchased from Transnetyx Tissue Inc and disassociated immediately using 2 mg/mL papain (Worthington Biochemical Corporation). Cells were initially cultured in media containing: Neurobasal media (Thermo Fisher Scientific), B-27 supplement (Thermo Fisher Scientific), and Glutamax supplement (Thermo Fisher Scientific), and 25 μM glutamate (Sigma Aldrich). 50% of the media was exchanged every 4 days with the fresh media containing no glutamate. Cells were plated onto 12 mm #1 round coverslips coated with Poly-D lysine and laminin (Corning).

## Flow cytometry-based FRET

Cells were transfected using polyethylenimine (PEI) 25 kDa linear polymer (Polysciences number 2396602). 1.5 μg of cerulean (Cer)- and venus (Ven)-tagged cDNA pairs were mixed together in 100 μl of serum-free DMEM media and 5 μl of PEI was added to each sterile tube. FRET experiments were performed two days post-transfection. The protein-synthesis inhibitor cycloheximide (100 μM) was added to cells 2 h before experimentation to halt synthesis of new fluorophores, in order to allow existing fluorophores to fully mature.

For FRET measurements, we used an LSR II (BD Biosciences) flow cytometer, equipped with 405 nm, 488 nm, and 633 nm lasers for excitation and 18 different emission channels. Forward- and side-scatter signals were detected and used to gate for single and healthy

cells. To determine FRET efficiency, we measured three distinct fluorescence signals: first, $S_{Cer}$ (corresponding to emission from the cerulean tag) is measured through the BV421 channel (excitation, 405 nm; emission, 450/50); second, $S_{Ven}$ (corresponding to emission from the venus tag) is measured via the FITC channel (excitation, 405 nm; dichroic, 505LP; emission, 525/50); and third, $S_{FRET}$ (corresponding to FRET emission) is measured via the BV510 channel (excitation, 405 nm; dichroic, 505LP; emission, 525/50). These raw fluorescence measurements are subsequently used to obtain $Ven_{direct}$ (venus emission due to direct excitation), $Cer_{direct}$ (cerulean emission due to direct excitation), and $Ven_{FRET}$ (venus emission due to FRET excitation). Fluorescence data were exported as FCS 3.0 files for further processing and analysis using custom MATLAB (2012b) functions (MathWorks).

For each experimental run on the flow cytometer, we performed several control experiments. First, the background fluorescence level for each fluorescent channel ($BG_{Cer}$, $BG_{Ven}$ and $BG_{FRET}$) was obtained by measuring fluorescence from cells exposed to PEI without any fluorophore-containing plasmids. Second, cells expressing the Ven fluorophore alone were used to measure the spectral crosstalk parameter $R_{A1}$, corresponding to bleed-through of Ven fluorescence into the FRET channel. Third, cells expressing the Cer fluorophore alone were used to measure spectral crosstalk parameters $R_{D1}$ and $R_{D2}$, corresponding to bleed-through of Cer fluorescence into the FRET and Ven channels respectively. Fourth, FRET measurements also require determination of instrument-specific calibration parameters $g_{Ven}/g_{Cer}$ and $f_{Ven}/f_{Cer}$, which are respectively ratios of fluorescence excitation and emission for Ven to Cer fluorophores. These parameters also incorporate fluorophore-dependent aspects, including molar extinction (for $g$) and quantum yield (for $f$), as well as instrument-specific parameters, including laser power, attenuation by optical components, and photodetection, amplification and digitization of fluorescence. To determine these parameters, we used Cer–Ven dimers with four different linker lengths (5, 32, 50 and 228). Fifth, co-expression of Cer and Ven fluorophores provided estimates for concentration-dependent collisional FRET.

In our experiments, $R_{A1}$ was approximately 0.11, $R_{D1}$ approximately 2.8, and $R_{D2}$ approximately 0.006. We observed only minor day-to-day variation in these measurements. For each cell, spectral cross-talk was subtracted as follows:

$$Cer_{direct} = R_{D1} \times S_{Cer} \quad (3)$$

$$Ven_{direct} = R_{A1} \times (S_{Ven} - R_{D2} \times S_{Cer}) \quad (4)$$

$$Ven_{FRET} = S_{FRET} - R_{A1} \times (S_{Ven} - R_{D2} \times S_{Cer}) - R_{D1} \times S_{Cer.} \quad (5)$$

Following spectral unmixing, we obtained $g_{Ven}/g_{Cer}$ and $f_{Ven}/f_{Cer}$ from data for Cer–Ven dimers by determining the slope and intercept for the following linear relationship:

$$\frac{Ven_{FRET}}{Cer_{direct}} = \frac{g_{Cer}}{g_{Ven}} \times \frac{Ven_{direct}}{Cer_{direct}} - \frac{f_{Ven}}{f_{Cer}} \quad (6)$$

For typical experiments, $g_{Ven}/g_{Cer} = 0.0194$ and $f_{Ven}/f_{Cer} = 2.3441$. Having obtained these calibration values, we computed donor-centric FRET efficiencies as:

$$E_D = \frac{Ven_{FRET}}{Ven_{FRET} + \frac{f_{Ven}}{f_{Cer}} \times Cer_{direct}} \quad (7)$$

For Cer–Ven dimers, we obtained FRET efficiencies of roughly 0.55, 0.38, and 0.05 for linker lengths 5, 32, and 228 respectively. The

relative proportion of Cer and Ven fluorophores in each cell was determined:

$$N_{Cer} = Cer_{direct}/(1 - E_D) \quad (8)$$

$$N_{Ven} = Ven_{direct}/(g_{Ven}/g_{Cer} \times f_{Ven}/f_{Cer}). \quad (9)$$ To construct FRET two-hybrid-binding curves, we imposed a 1:1 binding isotherm as in previous studies[29,76]. For each FRET pair, we obtained effective dissociation constant ($K_{d,EFF}$), $E_{D,max}$ and 95% confidence intervals by constrained least-squares fit.

## Flow cytometry assay of total and surface calcium channels

Cell surface and total ion channel pools were assayed by flow cytometry in live, transfected HEK293 cells as previously described[77]. Briefly, 48 h post-transfection, cells cultured in 12-well plates gently washed with ice cold PBS containing $Ca^{2+}$ and $Mg^{2+}$ (in mM: 0.9 $CaCl_2$, 0.49 $MgCl_2$, pH 7.4), and then incubated for 30 min in blocking medium (DMEM with 3% BSA) at 4 °C. HEK293 cells were then incubated with 1 μM Alexa Fluor 647 conjugated α-bungarotoxin ($BTX_{647}$; Life Technologies) in DMEM/3% BSA on a rocker at 4 °C for 1 h, followed by washing three times with PBS (containing $Ca^{2+}$ and $Mg^{2+}$). Cells were gently harvested in $Ca^{2+}$-free PBS, and assayed by flow cytometry using a BD Fortessa Cell Analyzer (BD Biosciences, San Jose, CA, USA) running BD FACSDiva (v8.65) acquisition software. CFP- and YFP-tagged proteins were excited at 407 and 488 nm, respectively, and Alexa Fluor 647 was excited at 633 nm.

## Electrophysiology

Whole-cell recordings of HEK293 cells were conducted 48 h after transfection using an EPC-10 patch clamp amplifier (HEKA Electronics) controlled by pulse software (HEKA Pulse v8.65). Micropipettes were prepared form 1.5 mm thin-walled glass (World Precision Instruments) using a P97 microelectrode puller (Sutter Instruments). Internal solution contained (mM): 135 cesium-methanesulfonate ($CsMeSO_3$), 5 CsCl, 5 EGTA, 1 $MgCl_2$, 2 MgATP, and 10 HEPES (pH 7.3). Series resistance was typically between 1–2 MΩ. There was no electronic resistance compensation. External solution contained (mM): 140 tetraethylammonium-$MeSO_3$, 5 $BaCl_2$, and 10 HEPES (pH 7.4). Whole-cell I-V curves were generated from a family of step depolarizations (−60 mV to +80 mV from a holding potential of −90 mV). Currents were sampled at 20 kHz and filtered at 5 kHz. Traces were acquired at a repetition interval of 10 s. Leak and capacitive transients were subtracted using a P/4 protocol.

Cell-attached single-channel recordings were performed at room temperature as described[37]. Patch pipettes (5–10 MΩ) were pulled from ultra-thick-walled borosilicate glass (BF200-116-10, Sutter Instruments), and coated with Sylgard. Currents were filtered at 2 kHz. The pipette solution contained 140 mM tetraethylammonium methanesulfonate; 10 mM HEPES; 40 mM $BaCl_2$; at 300 mOsm l$^{-1}$, adjusted with tetraethylammonium methanesulfonate; and pH 7.4, adjusted with tetraethylammonium hydroxide. To maintain the membrane potential at 0 mV, the bath contained 132 mM potassium glutamate, 5 mM KCl, 5 mM NaCl, 3 mM $MgCl_2$, 2 mM EGTA, 10 mM glucose, 20 mM HEPES; at 300 mOsm L$^{-1}$, adjusted with glucose; and pH 7.4, adjusted with sodium hydroxide. Cell-attached single-channel currents were measured during 200 ms voltage ramps between −80 mV and +70 mV (portions between −50 mV and +40 mV are displayed and analyzed). For each patch, we recorded 80–120 sweeps with a repetition interval of 10 s.

Whole-cell recordings of guinea pig cardiomyocytes were performed 48 h after infection, with internal solution comprised of (mM): 150 $CsMeSO_3$, 10 EGTA, 5 CsCl, $MgCl_2$, 4 MgATP, and 10 HEPES. For formation of gigaohm seals and initial break-in to the whole-cell configuration, cells were perfused in Tyrode solution containing (mM): 138 NaCl, 4 KCl, 2 $CaCl_2$, 1 $MgCl_2$, 0.33 $NaH_2PO_4$, and 10 HEPES (pH 7.4). Upon successful break-in, the perfusing media was switched to an

external solution composed of (mM): 155 *N*-methyl-D-glucamine, 10 4-amino-pyridine, 1 MgCl$_2$, 5 BaCl$_2$, and 10 HEPES (pH 7.4). Currents were sampled at 20 kHz and filtered at 5 kHz. Leak and capacitive transients were subtracted using a P/4 protocol.

## Immunofluorescence staining

Approximately 48 h after adenoviral infection guinea pig cardiomyocytes were fixed in 4% paraformaldehyde (wt/vol, in PBS) for 20 min at RT. Cells were washed twice with PBS and then incubated in 0.1 M glycine (in PBS) for 10 min at RT to block free aldehyde groups. Fixed cells were then permeabilized with 0.2% Triton X-100 (in PBS) for 20 min at RT. Non-specific binding was blocked with a 1 h incubation at RT in PBS solution containing 3% (vol vol$^{-1}$) normal goat serum (NGS), 1% BSA, and 0.1% Triton X-100. Cells were then incubated with rabbit anti-Ca$_V$1.2 primary antibody (Alomone Labs, 1:1000) in PBS containing 1% NGS, 1% BSA, and 0.1% BSA overnight at 4 °C. Cells were washed three times for 10 min each with PBS containing 0.1% Triton X-100 and then stained with anti-rabbit 488 secondary antibody (Thermofisher, 1:1000) for 1 h at RT. Antibody dilutions were prepared in PBS solution containing 1% NGS, 1% BSA, and 0.1% Triton X-100. The cells were then washed in PBS with 0.1% Triton X-100 and imaged in the same solution.

Hippocampal neurons were fixed with ice-cold 4% paraformaldehyde in phosphate buffered saline supplemented with 4% sucrose at room temperature for 10 min. Cells were then washed 3x in PBS, permeabilized with 0.2% Triton X-100 in PBS for 5 min at room temperature, and washed 3x in PBS. The cells were blocked at room temperature for 1 h in PBS + 1% BSA + 3% normal goat serum + 0.1% Triton X-100, and then incubated in primary antibody, diluted in PBS + 0.1% Triton X-100 + 1% BSA + 3% NGS overnight. Primary antibodies: anti-pCREB (1:333 dilution, Cell Signaling Technology), MAP2 (1:1000 dilution, Santa Cruz Biotechnology), Ca$_V$β$_1$ (1:200 dilution, Alomone). The following day, cells were washed 4x with PBS, then incubated in secondary antibody diluted in PBS + 0.1% Triton X-100 + 1% BSA + 3% NGS. Secondary antibodies: anti-rabbit Alexa-488, anti-mouse Alexa-647 (all 1:1000, Thermo Fisher Scientific). Hippocampal neurons were imaged using Nikon Eclipse Ti A1-A laser scanning confocal microscope with NIS-Elements AR 5.02.00 64-bit software.

## Pulldown assays

60 mm dishes of transfected HEK293 cells were harvested in PBS, centrifuged at 2000 g (4 °C) for 5 min and the pellet resuspended in NP40 lysis buffer containing (mM): 150 NaCl, 50 Tris (pH 8), 1% Triton X-100, and supplemented with protease inhibitor mixture (10 μL mL$^{-1}$, Sigma Aldrich), 1 PMSF, 2 N-ethylmaleimide, .05 PR-619 deubiquitinase inhibitor (LifeSensors). Cells were lysed on ice for 1 h with intermittent vortexing and centrifuged at 10,000 g for 10 min (4 °C). The soluble lysate was collected and protein concentration determined with the bis-cinchonic acid protein estimation kit (Pierce Technologies).

For Ca$_V$β$_1$ pulldowns, lysates were precleared with 10 μL of protein A/G sepharose beads (Rockland) for 1 h at 4 °C and then incubated with 2 μg anti-Ca$_V$β$_1$ antibody (UC Davis/NIH NeuroMab Facility, clone N7/18) for 1 h at 4 °C. Equivalent amounts of protein were then added to spin columns with 25 μL equilibrated protein A/G Sepharose beads and rotated overnight at 4 °C. Immunoprecipitates were washed a total of five times with NP40 buffer and then eluted with 30 μL elution buffer (50 mM Tris, 10% (vol vol$^{-1}$) glycerol, 2% SDS, 100 mM DTT, and 0.2 mg mL$^{-1}$ bromophenol blue) at 55 °C for 15 min. For Ca$_V$1.2 α$_{1C}$ pulldowns, lysates were added to spin columns containing 10 μL of equilibrated RFP-trap agarose beads, rotated at 4 °C for 1 h, and then washed/eluted as described above. Proteins were resolved on a 4–12% Bis Tris gradient precast gel (Life Technologies) in MOPS-SDS running buffer (Life Technologies) at 200 V constant for ~1 h. Protein bands were transferred by tank transfer onto a polyvinylidene difluoride (PVDF, EMD Millipore) membrane in transfer buffer (25 mM Tris pH 8.3, 192 mM glycine, 15% (vol/vol) methanol, and 0.1% SDS). The

membranes were blocked with a solution of 5% nonfat milk (BioRad) in tris-buffered saline-tween (TBS-T) (25 mM Tris pH 7.4, 150 mM NaCl, and 0.1% Tween-20) for 1 h at RT and then incubated overnight at 4 °C with primary antibodies (anti-FLAG HRP, Sigma Aldrich; Actin, Sigma Aldrich; α$_{1C}$, Alomone) in blocking solution. The blots were washed with TBS-T three times for 10 min each and then incubated with secondary horseradish peroxidase-conjugated antibody for 1 h at RT. After washing in TBS-T, the blots were developed with a chemiluminescent detection kit (Pierce Technologies) and then visualized on a gel imager. Membranes were then stripped with harsh stripping buffer (2% SDS, 62 mM Tris pH 6.8, 0.8% β-mercaptoethanol) at 50 °C for 30 min, rinsed under running water for 2 min, and washed with TBST (3x, 10 min). Membranes were pre-treated with 0.5% glutaraldehyde and re-blotted with anti-ubiquitin (VU1, LifeSensors, 1:500) as per the manufacturers' instructions.

## Western blot

Hippocampal cultures were harvested in PBS, centrifuged at 500 g for 5 min, then resuspended in NP40 lysis buffer (as above). 25 μg of protein per sample was loaded onto a PVDF membrane and probed as above using Cavβ$_1$, Cavβ$_2$, Cavβ$_3$ (Alomone, 1:1000), Cavβ$_4$ (NeuroMab, 1:1000) and actin (Sigma Aldrich, 1:1000).

## Skeletal muscle Ca$^{2+}$ imaging

Fiber loading with rhod-2 AM, a membrane-permeable non-ratiometric high affinity Ca$^{2+}$ indicator (Thermo Fisher, Cat. No. R1244), and subsequent imaging and analyses were performed as previously described[78,79] but with some modifications. Briefly, cultured FDB fibers were loaded with rhod-2 (2 μM for 60 min at 22 C) in 1 mL of L-15 media (ionic composition in mM: 137 NaCl, 5.7 KCl, 1.26 CaCl$_2$, 1.8 MgCl$_2$, pH 7.4; Life Technologies, Carlsbad, CA) supplemented with 0.25% w/v bovine serum albumin (BSA; Sigma-Aldrich, St Louis MO, Cat. No. A-7906). The fibers were washed thoroughly with appropriate L-15 media to remove residual fluorescent dye. All single fiber recordings were performed at room temperature. Confocal imaging of rhod-2 (100 μs/line) was performed using high-speed confocal system LSM 5 Live system (Carl Zeiss, Jena, Germany). Rhod-2 was excited with a 532 nm laser, and the fluorescence emitted >550 nm was detected on a Zeiss Axiovert 200 M inverted microscope and confocal imaging was performed in line scan *xt* mode as previously described[80], with images acquired for 0.4 to 1 s, using a 63 × 1.2 N.A. water immersion objective.

## Hippocampal calcium imaging

Hippocampal neurons were washed twice in basal solution containing (mM): 150 NaCl, 5 KCl, 2 CaCl$_2$, 2 MgCl$_2$, 10 HEPES, 10 D-glucose, pH 7.4, and incubated in the same solution containing 2 μM fura-2 with 0.05% Pluronic F-127 detergent (Life Technologies) for 30 min at 37 °C, 5% CO$_2$. Cells were subsequently washed twice in the same solution and placed on an inverted Nikon Ti-eclipse microscope with a Nikon Plan fluor 20x objective (0.45 N.A.). Fura-2 measurements were recorded at excitation wavelengths of 340 and 380 nm using EasyRatioPro (HORIBA Scientific). Hippocampal neurons were depolarized with a solution in which NaCl was reduced to 65 mM and KCl increased to 90 mM.

## Excitation-transcription coupling

Cultured hippocampal neurons were infected with lentivirus approximately 7 days after plating and used for experiments 7–10 days afterwards. Experiments were performed as previously described[81]. Cells were pre-incubated for 4 h in culture media supplemented with: 10 μM NBQX (Tocris Biosciences), 10 μM AP5 (Tocris Biosciences), and 0.5 μM tetrodotoxin (Tocris Biosciences). All solutions used throughout the experiment contained these blockers. Cells were then washed twice in hippocampal basal imaging solution (as above), and incubated for 5 min in 5 K basal solution (+ any calcium channel blockers). Cells were then placed in 90 K solution (plus indicated calcium channel

blockers) for 3 min and immediately fixed with ice-cold 4% paraformaldehyde in phosphate buffered saline supplemented with 4% sucrose. Samples were immediately processed for immunostaining, as described above.

## Protein expression and purification

Cav$\beta_{2a}$-link[34], a construct of rat Cav$\beta_{2a}$ (GenBank NM_053851) consisting of the SH3 domain (residues 17–138) and the NK domain (residues 203–425) separated by a serine residue[34], a construct covering rat Cav$\beta_{1b}$ (GenBank NM_059042.2) residues 58 to 427, nb.F3, and nb.E8 were expressed using pET28HMT[31], a vector containing in series a hexahistidine tag, maltose binding protein (MBP), and Tobacco Etch Virus (TEV) protease site N-terminal to the protein of interest. Cav$\beta_{1b}$ nb.F3, and nb.E8 were cloned into this vector by Gibson assembly[64].

Cav$\beta_{2a}$-Link was expressed and purified as described previously[34]. For Cav$\beta_{1b}$−58-427, 100 ng of plasmid was transformed into Rosetta™(DE3)pLysS competent cells, plated on LB (Luria Broth) agar containing 50 µg ml$^{-1}$ Kanamycin and 34 µg ml$^{-1}$ Chloramphenicol, and grown overnight at 37 °C. A single colony was used to inoculate 60 ml LB media containing 50 µg ml$^{-1}$ Kanamycin and 34 µg ml$^{-1}$ Chloramphenicol and was grown overnight at 37 °C. Six 1 L flasks containing 2x YT media (5 g NaCl, 16 g tryptone, 10 g yeast extract) supplemented with 50 µg ml$^{-1}$ Kanamycin and 34 µg ml$^{-1}$ Chloramphenicol were each inoculated with 10 ml of overnight starter culture per liter and grown at 37 °C in a shaker incubator until OD$_{600}$ reached 0.4, at which point the temperature was changed to 18 °C and cultures were induced with 1 mM isopropyl β-d-thiogalactopyranoside (IPTG) after 30 minutes and then grown overnight. Cells were harvested by centrifugation (4000 g × 20 min). Cell pellets were flash frozen in liquid nitrogen and stored at −80 °C until further use. All protein purification steps, unless mentioned otherwise, were carried out at 4 °C. 6 L worth of bacterial pellets were suspended in 100 ml of ice-cold lysis buffer containing 200 mM KCl, 20 mM Imidazole, 10% glycerol, 1 mM Phenylmethylsulphonyl fluoride (PMSF), 0.1 mg ml$^{-1}$ DNaseI, 10 mM HEPES/KOH, pH 7.4 and lysed using an EmulsiFlex-C5 homogenizer (Avestin) (2 passages, 5000 psi). Lysate was clarified by ultracentrifugation at 100,000 g for 1 hour at 4 °C. Supernatant was applied to a 20 ml Poros 20 MC Ni$^{2+}$ column (Applied Biosystems) and equilibrated with Buffer A (200 mM KCl, 20 mM Imidazole, 10% glycerol, 10 mM HEPES/KOH pH 7.4). The column was washed with two column volumes (CVs) of Buffer A, followed by two CVs of 94% Buffer A and 6% Buffer B (200 mM KCl, 500 mM Imidazole, 10% glycerol, 10 mM HEPES/KOH pH 7.4). Bound protein was eluted using two CVs of 40% Buffer A and 60% Buffer B. Eluted protein was concentrated to <1 ml using centrifugal concentrator (Amicon filter, MWCO 50 kDa) and then diluted to 50 ml in TEV digestion buffer (200 mM KCl, 10 mM β-Mercaptoethanol (BME) 10 mM HEPES/KOH pH 7.4). Affinity tag (Histidine tagged MBP) was removed by addition of 50 µl of 1 mg ml$^{-1}$ 6x histidine tagged superTEV[82], rotating at room temperature overnight. Cleaved protein was separated from the affinity tag and protease by passing it through POROS 20MC Ni$^{2+}$ column (Applied Biosystems), equilibrated with Buffer A. Flowthrough was collected, concentrated to <1 ml using a centrifugal concentrator (Amicon filter, MWCO 30 kDa), diluted to 50 ml in buffer B50 (50 mM KCl, 20 mM HEPES/KOH pH 7.4, 10 mM BME), and applied to a 10 ml Hiload SP HP column (Cytiva Life Sciences) for ion exchange chromatography. Protein was eluted using a linear gradient of 0-100% buffer B1000 (1000 mM KCl, 20 mM HEPES/KOH pH 7.4, 10 mM BME) over 20 CVs. Purity was validated using SDS-PAGE. Fractions containing highest purity were pooled and concentrated to >30 mg ml$^{-1}$ using a centrifugal concentrator (Amicon filter, MWCO 30 kDa), flash frozen in liquid nitrogen, and stored at −80 °C.

Nb.F3 and nb.E8 were expressed and purified as follows: 100 ng plasmid was transformed into Shuffle™ competent cells and plated on LB agar plates containing 50 µl ml$^{-1}$ Kanamycin and overnight a 30 °C.

60 ml LB containing 50 µl ml$^{-1}$ Kanamycin was inoculated using a single colony and grown overnight at 30 °C. Six 1 L flasks of 2x YT media containing 50 µl ml$^{-1}$ Kanamycin were each inoculated with 10 ml of starter culture and grown at 30 °C until OD$_{600}$ reached 0.4. Temperature of the shaker was then reduced to 18 °C and cultures were induced at 0.6 OD$_{600}$ with 1 mM IPTG and grown overnight. Cells were harvested by centrifugation at 4000 g for 20 min. Bacterial pellets were flash frozen in liquid nitrogen and stored at −80 °C until further use. All protein purification steps, unless mentioned otherwise, were carried out at 4 °C. 6 L bacterial pellet was resuspended in 100 ml ice-cold lysis buffer. Resuspended cells were lysed using emulsiflex (2 passages, 5000 psi) and clarified by ultracentrifugation at 100,000 g for 1 h at 4 °C. Supernatant was applied to 20 ml Poros 20 MC Ni$^{2+}$ column equilibrated with Buffer A. Column was washed with 2 CV Buffer A followed by 2 CV of 94% Buffer A and 6% Buffer B before elution with 2 CV of 40% Buffer A and 60% Buffer B. 10 ml fractions were collected and protein was concentrated to <1 ml using centrifugal concentrator (Amicon filter, MWCO 30 kDa). Eluted protein was concentrated to <1 ml using a centrifugal concentrator (Amicon filter, MWCO 10 kDa) and then diluted to 50 ml in TEV digestion buffer (200 mM KCl, 10 mM HEPES/KOH pH 7.4). Affinity tag (Histidine tagged MBP) was removed by addition of 50 µl of 1 mg ml$^{-1}$ 6x histidine tagged superTEV[82], rotating at room temperature overnight. Cleaved protein was separated from the affinity tag and protease by passing it through POROS20MC Ni$^{2+}$ column (Applied Biosystems), equilibrated with Buffer A. Flowthrough was collected, concentrated to <1 ml using centrifugal concentrator (Amicon filter, MWCO 10 kDa), and purified on a 24 ml Superdex75 10/300 GL (GE Healthcare) column equilibrated with SEC buffer (150 mM KCl, 10% Glycerol, 10 mM HEPES/KOH pH 7.4). Purity was validated using SDS-PAGE. Fractions with highest protein purity were pooled and concentrated to >30 mg ml$^{-1}$ using a centrifugal concentrator (Amicon filter, MWCO 10 kDa), flash frozen in liquid nitrogen, and stored at −80 °C.

## Protein crystallization and structure determination

For Cav$\beta_{2a}$-Link:nbF3 complex formation, 50 nmoles of Cav$\beta_{2a}$-Link (Molecular weight (MW)− 39.4 kDa) was incubated with 100 nmoles of nbF3 (MW- 13.78 kDa) (1:2 molar ratio) in SEC buffer with final reaction volume of 500 µl for 30 min at 4 °C. For Cav$\beta_{1b}$−58-427:nbF3:nbE8 complex formation, 50 nmoles of Cav$\beta_{1b}$−58-427 (MW- 41.24 kDa) was combined with 100 nmoles of nb.F3 and 100 nmoles of nb.E8 (MW-14.3 kDa) in 1:2:2 molar ratio, in total SEC buffer with final 500 µl reaction volume. The proteins were incubated for 30 min at 4 °C. Nanobody complexes were isolated by injecting the incubated protein mixes onto a 24 ml Superdex200 10/300 GL column (Cytiva Life Sciences) equilibrated with SEC buffer. Complex formation was validated by SDS PAGE, concentrated to ~10 mg ml$^{-1}$ using centrifugal concentrator (Amicon filter, MWCO 3 kDa), and used immediately for crystallization by sitting drop vapor diffusion at 4 °C using 1:1 ratios of protein:precipitant. Protein concentrations were determined by absorbance at 280 nm[83].

The Cav$\beta_{2a}$-Link:nbF3 complex crystallized in 30–40% MPD, 0.1 M HEPES/KOH pH 7.0. Crystals were harvested at 4 °C using 40% MPD as cryoprotectant. The Cav$\beta_{1b}$−58-427:nb.F3:nb.E8 complex crystallized in 15–22% PEG3350, 0.15–0.25 M Ammonium Sulphate. Crystals were cryoprotected by gradual increment of glycerol concentration, starting from 5% to 30%, in mother liquor. Finally, the crystals were harvested with 30% glycerol as cryoprotectant.

Diffraction data were collected at 100 K at Beamline 8.3.1 (Advanced Light Source, Lawrence Berkeley National laboratories) and APS GM/CAT beamline 23-IDB/D Chicago, Illinois. Data was indexed using XDS[84] and scaled and merged using AIMLESS[85]. Molecular replacement was done in PHASER[86] using PDB:5V2P chain A[35]. Model was built and improved by successive rounds of building and refinement using COOT[87] and Phenix[88], respectively.

## Data and statistical analysis

Data were analyzed off-line using FloJo, PulseFit, Microsoft Excel, Origin and GraphPad Prism software. Statistical analyses were performed in Origin, Microsoft Excel or GraphPad Prism using built-in functions. Statistically significant differences between means ($P < 0.05$) were determined using Student's $t$ test for comparisons between two groups or one-way ANOVA for multiple groups, with Tukey's post-hoc analysis. Data are presented as means ± SEM.

## Reporting summary

Further information on research design is available in the Nature Portfolio Reporting Summary linked to this article.

## Data availability

The data that support this study are available from the corresponding authors upon reasonable request. Coordinates and structure factors have been deposited in the Protein Data Bank (PDB) under accession codes 8DAM (Ca$_V$β$_{1b}$:nb.E8) and 8E0E (Ca$_V$β$_{2a}$:nb.F3). Previously published PDB can be accessed via accession code 5V2P [https://doi.org/10.2210/pdb7VF9/pdb] (Ca$_V$β$_{2a}$:Ca$_V$1.2 AID peptide complex). The source data underlying Fig. 2a, b, i, Fig. 4b, c, e, f, h, i, k, l, Fig. 5c, g, Fig. 6c-f, h-k, Fig. 7a–c, Fig. 8c, d, g, h, Fig. 9b, d, f, Supplementary Fig. 5, Supplementary Fig. 7b, and Supplementary Fig. 8b are provided as a Source Data File. Source data are provided with this paper.

## Code availability

Flow-FRET and single channel recordings were analyzed using custom Matlab (2012b) software. The code is accessible at: https://github.com/manubenjohny/FACS_FRET and https://github.com/manubenjohny/SinglesAnalysis/, respectively.

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

## Acknowledgements

We thank Ming Chen for technical support. This work was supported by grants RO1-HL121253 and RO1-HL122421 from the NIH (to HMC), F31 DK118866 (to T.J.M.), R01-HL080050 (to D.L.M.), and R01-AR075726 and R01-NS103777 to (M.F.S.). Flow cytometry experiments were performed in the CCTI Flow Cytometry Core, supported in part by the NIH (S10RR027050). Confocal images were collected in the HICCC Confocal and Specialized Microscopy Shared Resource, supported by NIH (P30 CA013696). This work was prepared while R.A.B. was employed at the University of Maryland Baltimore. The opinions expressed in this article are the author's own and do not reflect the view of the National Institutes of Health, the Department of Health and Human Services or the United States government.

## Author contributions

T.J.M. and H.M.C. conceived the study; T.J.M., N.N., E.H.-O., M.B.J., R.A.B., M.F.S., D.L.M., H.M.C. designed the experiments; T.J.M., E.H.-O., and J.D.L. performed electrophysiology and Ca²⁺ imaging experiments; N.N. and D.L.M. performed and analyzed structural biology experiments; P.C. and T.J.M. performed FLOW-fret experiments; M.B.J. analyzed flow-fret and single-channel analyses; E.H.-O., H.B., and R.A.B. performed skeletal muscle experiments; T.J.M., N.N., E.H.-O., R.A.B., M.F.S., D.L.M., H.M.C. analyzed data; T.J.M., H.M.C. wrote the paper; T.J.M., N.N., E.H.-O., J.D.L., R.A.B., M.S., D.L.M., H.M.C. edited the paper; T.J.M., D.L.M., H.M.C., M.F.S. obtained funding.

## Competing interests

T.J.M. and H.M.C. have filed a patent application through Columbia University based on this work. T.J.M. and H.M.C., 2019. Composition and methods for genetically-encoded high voltage-activated calcium channel blockers using engineered ubiquitin ligases (U.S. Application Serial No. 62/830,142). The remaining authors declare no competing interests.
