## [Peer Review File · Nature Communications]

Selective posttranslational inhibition of CaV β 1-associated voltage-dependent calcium channels with a functionalized nanobodyReviewers' Comments:

Reviewer #1:

Remarks to the Author:

In this study, Morgenstern et al describe the generation of nanobody that binds to Cavbeta1-associated Ca channels and thus decreases the channel's open probability as well as its trafficking to the membrane. Overall, I found the paper to outstanding for its novelty, rigorous experimentation and analyses, and broad impact. Although, in my view, additional experiments are not required, changes to the discussion and results sections are important to improve the manuscript. I list these comments below.

1) At this point, after so many trafficking papers, the term "trafficking" is too broad and ambiguous. For any membrane protein, it encompasses insertion, membrane dwell time, removal and, potentially, recycling. The paper by Sato et al 2019 offers a framework on how to discuss how nb.E8 could alter channels "trafficking". This may be important in the discussion of how anything may alter "trafficking". For example, while at any particular time the number of functional number of calcium channels in a ventricular myocyte remains about the same, their dwell time in the membrane may be greatly reduced. This is a trafficking problem, but may not be seen as one if one only looks at snapshots of protein distribution or even currents. The authors should modify their discussion to refine their model.

2) I am a bit ambivalent of Figure 1 and its lengthy discussion in the main section the paper. While it makes for good story telling, it does not seem to add new information to the paper.

Reviewer #2:

Remarks to the Author:

The authors identified a nanobody nb.E8 and demonstrate, in a rigorous set of detailed experiments, that nb.E8 selectively binds (in a post-translational manner) the CavB1 SH3 domain and inhibits CavB1 associated HVACCs. The finding is an important following on an earlier report (reference 28) in which another nanobody was found to non-selectively inhibit HVACC beta subunits. The study introduces an important new tool to study calcium channel biology and their host organs including, skeletal myocytes, cardiac myocytes, and neurons. The ability for nb.E8 to selectively inhibit calcium current is impressive and could be further improved if data are provided to better understand the mechanisms by which nb.E8 / Chisel-1 affects calcium channel trafficking and gating, as discussed in specific comments below.

1. To understand the mechanism of action of Chisel-1, it would be helpful to understand when it binds Cavbeta1 – prior or after Cavbeta1 associated with the alpha unit? Does Chisel-1 limit the alpha subunit trafficking to the membrane or increase internalization from the membrane?

2. Why are Po and gating of HVACCs affected by Chisel-1? Can the authors quantify the relative effect of Chisel-1 on N (i.e. trafficking and surface expression) versus gating?

3. Is ubiquitination and degradation the mechanism by which Chisel decreases the amount of beta1 and alpha1 protein (Figure 6)?

4. The amount of protein degradation of beta1 and alpha1 is much more limited than the almost complete loss of current (Figure 6h and 7c). Similarly, loss of surface channel is much more modest than an almost complete loss of current (Figure 5c). The impressive ability of nb.E8 and Chisel-1 to almost completely eliminate HVACC current is not understood.

5. How was Chisel-1 introduced into isolated adult guinea-pig cardiomyocytes?

6. The skeletal versus cardiac specificity data (Figure 7) are impressive. The cardiac experiment would benefit from a positive control. Can Cav-aBlator be used in the cardiomyocytes to eliminate Cav1.2 current?

Reviewer #3:

Remarks to the Author:

For three decades calcium channel researchers invoked the potential of the specific alpha – beta interactions as potential means for selectively modulating or inhibiting CaV channels in specific tissues. In their recent work, including this article, Henry Colecraft and collaborators successfully devised a strategy how this can be achieved. While alpha subunit specific inhibitors are available for some CaV channels, the auxiliary beta subunits have distinct expression profiles. Thus, specifically targeting beta subunits with an isoform-specific nanobody opens new possibilities to generate inhibitors with a very distinct pharmacological profile, as exemplified here by the selective blocking of L-type calcium channel-specific functions in neurons and in skeletal muscle, but not in the heart. This work is highly innovative, of great interest to the calcium channel community and beyond, as the experimental strategy may well be applicable to a wide range of channels and receptors.

The authors isolated a b1 specific nanobody from their previously established phagemid library and characterized its subunit specificity and the binding epitope in the b1 SH3 domain using a cellular FRET assay. Next, they characterized the exact binding sites of the pan-specific nb.F3 and the b1-specific nb.E8 at atomic resolution using X-ray crystallography. These experiments reveal the structural basis for the subunit-specificity of nb binding. Then the authors use electrophysiology to establish the functional selectivity of nb.E8 relative to the beta and alpha isoforms. As expected from the loss of beta function, the observed reduction of the current density was explained by a combination of reduced surface expression (examined with a BTX binding assay) and reduced open probability (shown in single channel recordings). To boost the efficacy of their b1-specific nb they fused the HECT domain of Nedd4L to nb.E8, as previously done with nb.F3, which totally abolished CaV1.3, CaV2.2, and CaV2.3 currents in b1 expressing HEK cells, while maintaining the specificity for b1. Finally, the authors tested the efficacy of their b1-specific inhibitor in native muscle and nerve cells. First, they demonstrate the specific ablation of L-type calcium currents in skeletal muscle fibers transfected with Chisel-1, but not in cardiac myocytes, in which the L-type calcium channel associates with b2. Also, forskolin-induced enhancement of the cardiac calcium currents, which is dependent on the CaV1.2 – beta interaction, remained fully intact. In the last set of experiments, the authors show that selective block of b1 with Chisel-1 partially blocks the nuclear translocation of pCREB in response to strong K⁺ depolarization, thus showing that Chisel-1 also works in an environment where multiple CaV alpha and beta subunits are simultaneously expressed.

This characterization of the nb.E8 and Chisel-1 is very complete, the methods are state of the art, the data and the figures convincing and clearly presented. The differential effects on skeletal and cardiac L-type calcium channels impressively demonstrate the selectivity of Chisel-1 and the experiment in the hippocampal neurons provides an outlook of what might be possible. The data fully support the conclusion reached in the article. Overall, the article is well written. It was a pleasure to read and there is little, if anything, to criticize.

Comments

The experiments shown in supplementary figures 6 and 7 are of central importance for the interpretation of the following results. Therefore, these figures should preferentially be displayed in the primary article.

Figure 6. In the beta WBs the multiple bands should be labeled for clarity.

Figure 7. Since the physiological role of CaV1.1 in skeletal muscle fibers is depolarization-induced calcium release (excitation-contraction coupling), it would be very interesting to see the effect of Chisel-1 on depolarization-induced calcium transients in the muscle fibers. Particularly because there is full block of ICa, whereas charge movement is only reduced by about 50%, it is important to see to what degree ECC will be affected.

Figure 8b. It is unexpected that untreated neurons should express so little beta4. Was this consistently observed? In the figures showing WBs, immunofluorescence or traces (particularly when without quantification; e.g. 8a,b,c; 7d) please indicate in the legend that these are representative examples and how many repeats have been performed.

Figure 8. Optional, but still very interesting would be the direct comparison of the Chisel-1 effects with that of the corresponding non-selective beta inhibitor on nuclear pCREB.

Supplementary figure 9. Without showing appropriate controls and high-resolution images these immunofluorescence images are largely meaningless. As the expression and subcellular distribution patterns of alpha and beta subunits in hippocampal cultures are well known, it would be more appropriate to cite the relevant literature, rather than showing incomplete (and inferior) examples of the author's own repetitions of these experiments. Note that the shown neurons do not appear to express dendritic spines (the postsynaptic structure where some of the beta isoforms should be accumulated), indicating that the neurons are poorly differentiated. Also, how shall panel C be interpreted? Is CaV2.1 expression suppressed or its distribution pattern altered in the presence of Chisel-1? This figure raises more questions than answers. Better remove it; it is not critical for the study.

The primary antibodies used to label b2, b3, and b4 in figure S9 are not reported in the methods (IF staining of hippocampal neurons).

Sometimes the authors get a bit carried away in their use of flowery language that seems inappropriate for a scientific article. E.g. in line 70 "myriad of diseases"

Line 354 the hippocampal neuron figure is S9 not S5

Response to Reviewers

We thank the reviewers for their kind comments on the manuscript and for their constructive critiques. We have revised the manuscript to address all the reviewers' concerns.

Response to Reviewer #1:

1) At this point, after so many trafficking papers, the term "trafficking" is too broad and ambiguous. For any membrane protein, it encompasses insertion, membrane dwell time, removal and, potentially, recycling. The paper by Sato et al 2019 offers a framework on how to discuss how nb.E8 could alter channels "trafficking". This may be important in the discussion of how anything may alter "trafficking". For example, while at any particular time the number of functional number of calcium channels in a ventricular myocyte remains about the same, their dwell time in the membrane may be greatly reduced. This is a trafficking problem, but may not be seen as one if one only looks at snapshots of protein distribution or even currents. The authors should modify their discussion to refine their model.

We agree with the reviewer that the term "trafficking" may be too broad a description for the intended usage in the manuscript. We used "trafficking" seven times in the manuscript and have replaced all those mentions with either "surface density" or "targeting" which are more specific for the context used.

2) I am a bit ambivalent of Figure 1 and its lengthy discussion in the main section the paper. While it makes for good story telling, it does not seem to add new information to the paper.

We appreciate this feedback from the reviewer but have elected to keep Figure 1 in its current form. We believe that it is important to the story to be explicit up front about how and why the posttranslational knockdown approach is different and complementary to conventional gene knockout/knockdown methods. In presentations of the work, we have found that the different outcomes between the two approaches are underappreciated or unclear to audiences if not explained.

Response to Reviewer #2:

1. To understand the mechanism of action of Chisel-1, it would be helpful to understand when it binds Cavbeta1 – prior or after Cavbeta1 associated with the alpha unit? Does Chisel-1 limit the alpha subunit trafficking to the membrane or increase internalization from the membrane?

We agree with the reviewer that understanding how precisely Chisel-1 decreases CaV channel surface density is an interesting question, but one that we believe is outside the scope of the present manuscript. The totality of our results suggest that Chisel-1 forms

a ternary complex with the holo-channel. It is not clear to us how one could distinguish whether Chisel-1 binds to Cavbeta1 prior to or after it associates with the alpha subunit and how that would help understand the mechanism of action (given that a ternary complex is formed).

2. Why are P_o and gating of HVACCs affected by Chisel-1? Can the authors quantify the relative effect of Chisel-1 on N (i.e. trafficking and surface expression) versus gating?

Chisel-1 eliminates whole-cell current so we cannot explicitly measure effects on gating and P_o . In skeletal muscle, gating currents indicate that Chisel-1 reduces CaV1.1 surface density by 50% (Figure 8), while in HEK cells our flow cytometry assay indicates a 60% reduction of channels at the surface (Supplemental Figure 6). Thus, we infer from the near complete loss of whole-cell current that Chisel-1 has an even stronger effect on reducing the P_o of surface channels than we measured for nb.E8 (in Fig. 5). This could be due in part to the enhanced ubiquitination which has been reported to affect gating in some ion channels (e.g. Aisenberg et al 2022, JBC, 298:101826). We interpret the impact of nb.E8 on decreasing channel P_o as being due to its forming a ternary complex with the holo-channel via binding Cavbeta1 SH3 domain. It is known that Cavbeta binding to Cav channels via the NK/AID interaction regulates channel P_o . We speculate that nb.E8 (but not nb.F3) binds in a way that affects Cavbeta regulation of P_o . The finding that a protein binding to Cavbeta can exert effects on Cav channel P_o and surface density is not unprecedented and is in fact a phenomenon displayed by RGK proteins, for example.

We believe the data in Fig. 5 addresses the reviewer's question in relation to nb.E8; we observe a ~50% reduction in surface expression of the channel compared to control when we co-express nb.E8 in our flow cytometry BBS-labeling assay, as well as a 50% reduction in P_o from single channel recordings of those channels that do make it to the cell surface.

We address the discrepancy between Chisel-1 impact on surface density and whole-cell current in the revised manuscript (lines 341-345):

“The discrepancy between the near complete ablation of whole-cell current and the 50% reduction in Cav1.1 surface density suggests that Chisel-1 inhibits the P_o of channels remaining at the surface even more strongly than we found for nb.E8. This could be due to the enhanced ubiquitination which has been shown to inhibit gating of some channels ³⁹.”

3. Is ubiquitination and degradation the mechanism by which Chisel decreases the amount of beta1 and alpha1 protein (Figure 6)?

Ubiquitination is widely known to be a posttranslational modification that results in protein degradation. However, it is also known that not all ubiquitin modifications result in protein degradation and can have other signaling functions. We observe in Fig. 6 that Chisel-1 increases ubiquitination and degradation of both beta1 and alpha1. We, therefore,

assume Chisel-1 promotes degradation by enhancing ubiquitination. Nevertheless, we cannot be absolutely certain of this given the known non-degradative functions of some ubiquitin chains. We have modified the text to reflect this as follows (lines 302-304):

“When co-expressed with α_{1C} + β_{1b} , Chisel-1 decreased expression of both β_{1b} and α_{1C} , most likely due to ubiquitin-mediated degradation, as supported by the increased ubiquitination of both subunits”

4. The amount of protein degradation of beta1 and alpha1 is much more limited than the almost complete loss of current (Figure 6h and 7c). Similarly, loss of surface channel is much more modest than an almost complete loss of current (Figure 5c). The impressive ability of nb.E8 and Chisel-1 to almost completely eliminate HVACC current is not understood.

The reviewer is correct in noting that the degradation of beta1 and alpha1 by Chisel-1 is more limited than the complete loss of current observed in Figures 6h and 7c. We have previously shown that targeted ubiquitination of K⁺ channels and Cav channels can essentially eliminate currents even when protein degradation is modest or absent (Kanner et al, 2017, eLife; Morgenstern et al, 2019, eLife). This is because ubiquitination of the channels also leads to their intracellular retention. In cardiomyocytes, nb.F3 (pan CaVbeta nanobody) fused to Nedd4L HECT domain, completely eliminated Cav1.2 current without degrading alpha1C by trapping the channel in Rab7-positive late endosomes (Morgenstern et al., 2019, eLife). Thus, the effects of Chisel-1 are due to a combination of decreased surface expression (Supplemental Figure 6 in revised manuscript) due to intracellular retention and degradation, as well as diminished P_o of any channels that do remain on the cell surface (see response to point #2 above). We refer to our previous results in the revised text (lines 125-128).

“Expression of Cav- $\alpha\beta$ lator in cardiac myocytes eliminated $I_{Ca,L}$ by retaining Cav1.2 in intracellular organelles, primarily late endosomes ²⁸, definitively revealing that essentially all Cav1.2 α_{1C} subunits are bound to Cav β in adult ventricular cardiomyocytes (Fig. 1a).”

Also (Lines 349-351): **“Importantly, we previously showed that Cav- $\alpha\beta$ lator (comprised of nb.F3 fused to Nedd4L HECT domain) eliminates Cav1.2 current in adult guinea pig ventricular cardiomyocytes ²⁸.”**

Nb.E8 reduces whole-cell current by 78% (Figure 4a,b) which is accounted for by the combination of 50% decrease in surface density and ~50% decrease in P_o. Chisel-1 further depresses current by diminishing channel surface density even more (Supplemental Figure 6 in the revised manuscript).

5. How was Chisel-1 introduced into isolated adult guinea-pig cardiomyocytes?

We thank the reviewer for catching this omission. We have modified the text to address this omission as follows (Lines 346-348):

“In sharp contrast with the elimination of Cav1.1 current in skeletal muscle, adenovirus-mediated expression of Chisel-1 in adult guinea pig ventricular cardiomyocytes had no effect on whole-cell Cav1.2 functional expression”

6. The skeletal versus cardiac specificity data (Figure 7) are impressive. The cardiac experiment would benefit from a positive control. Can Cav-aBlator be used in the cardiomyocytes to eliminate Cav1.2 current?

We have cited our previous work indicating that Cav-ablator eliminates CaV1.2 current in cardiomyocytes (Lines 125-126):

“Expression of Cav-aβlator in cardiac myocytes eliminated $I_{Ca,L}$ by retaining Cav1.2 in intracellular organelles, primarily late endosomes²⁸...”

Also (Lines 349-351): **“Importantly, we previously showed that Cav-aβlator (comprised of nb.F3 fused to Nedd4L HECT domain) eliminates Cav1.2 current in adult guinea pig ventricular cardiomyocytes²⁸.”**

Response to Reviewer #3:

1. The experiments shown in supplementary figures 6 and 7 are of central importance for the interpretation of the following results. Therefore, these figures should preferentially be displayed in the primary article.

We have condensed Supplementary figures 6 and 7 into one new Figure which is now included in the primary article (Figure 7 in the revised manuscript) as the reviewer suggests.

2. Figure 6. In the beta WBs the multiple bands should be labeled for clarity.

Thank you. The bands in the figure have now been labeled with MW markers for clarity.

3. Figure 7. Since the physiological role of CaV1.1 in skeletal muscle fibers is depolarization-induced calcium release (excitation-contraction coupling), it would be very interesting to see the effect of Chisel-1 on depolarization-induced calcium transients in the muscle fibers. Particularly because there is full block of I_{Ca} , whereas charge movement is only reduced by about 50%, it is important to see to what degree ECC will be affected.

The reviewer is correct in noting that while there is full block of I_{Ca} in skeletal muscle fibers, the charge movement is only reduced by 50%. We have investigated the impact of Chisel-1 on depolarization-induced calcium transients in muscle fibers as the reviewer

suggested. We found that Chisel-1 decreased calcium transient amplitude by 50% as would be expected given that skeletal EC coupling is driven by voltage-induced Ca^{2+} release from the sarcoplasmic reticulum. This new result is now included as Supplemental Figure 7 and discussed in the revised manuscript (lines 335-339):

“Chisel-1 also reduced stimulus-evoked rhod2 Ca^{2+} transients by 50% compared to control skeletal muscle fibers expressing CFP (Supplemental Fig. 7s), consistent with excitation-contraction coupling in skeletal muscle being mediated by voltage-induced Ca^{2+} release rather than the Ca^{2+} -induced Ca^{2+} release found in heart cardiomyocytes³⁹.”

4. It is unexpected that untreated neurons should express so little beta4. Was this consistently observed? In the figures showing WBs, immunofluorescence or traces (particularly when without quantification; e.g. 8a,b,c; 7d) please indicate in the legend that these are representative examples and how many repeats have been performed.

Our results do show that beta4 is expressed in the untreated neurons; it just increases with Chisel-1, concomitant with a decrease in beta1 expression. We have indicated in the figure legend that the blots are representative of three independent repeats.

5. Optional, but still very interesting would be the direct comparison of the Chisel-1 effects with that of the corresponding non-selective beta inhibitor on nuclear pCREB.

We have now included this data in Figure 9. As expected, CaV-ablator inhibits excitation-induced phosphoCREB signal in the nucleus of hippocampal neurons to a similar extent as the cocktail of Cav channel blockers. The data is presented in lines 385-387:

“Cav- α blator inhibited excitation-induced phosphoCREB signal in the nucleus of hippocampal neurons to a similar extent as the cocktail of Cav channel blockers (Fig. 9f).”

6. Supplementary figure 9. Without showing appropriate controls and high-resolution images these immunofluorescence images are largely meaningless. As the expression and subcellular distribution patterns of alpha and beta subunits in hippocampal cultures are well known, it would be more appropriate to cite the relevant literature, rather than showing incomplete (and inferior) examples of the author’s own repetitions of these experiments. Note that the shown neurons do not appear to express dendritic spines (the postsynaptic structure where some of the beta isoforms should be accumulated), indicating that the neurons are poorly differentiated. Also, how shall panel C be interpreted? Is CaV2.1 expression suppressed or its distribution pattern altered in the presence of Chisel-1? This figure raises more questions than answers. Better remove it; it is not critical for the study.

We have followed the reviewer’s suggestion and removed Supplementary Figure 9 and cited relevant literature showing this information (lines 363-364):

“Hippocampal neurons express multiple Cav channel α_1 (including Cav1.2, Cav1.3, Cav2.1, Cav2.2, and Cav2.3) and Cav β (Cav β_1 -Cav β_4) subunits ⁴¹⁻⁴⁵,...”

7. The primary antibodies used to label b2, b3, and b4 in figure S9 are not reported in the methods (IF staining of hippocampal neurons).

Thank you. We have indicated the source of antibodies in the Methods section (**lines 905-907**):

“25 μ g of protein per sample was loaded onto a PVDF membrane and probed as above using Cav β_1 , Cav β_2 , Cav β_3 (Alomone, 1:1000), Cav β_4 (NeuroMab, 1:1000) and actin (Sigma Aldrich, 1:1000).”

8. Line 354 the hippocampal neuron figure is S9 not S5.

Thank you. Figure S9 was removed in agreement with the reviewer's point #6 above.

Reviewers' Comments:

Reviewer #1:

Remarks to the Author:

The authors addressed all concerns. Congratulations for a fantastic paper.

Reviewer #2:

Remarks to the Author:

Detailed responses appreciated, no further concerns.

Reviewer #3:

Remarks to the Author:

The authors have addressed all of my previous concerns and I recommend publication of the article in its present form.